# High-fat diet fuels prostate cancer progression by rewiring the metabolome and amplifying the MYC program

David P. Labbé 🆔 et al.#

Systemic metabolic alterations associated with increased consumption of saturated fat and obesity are linked with increased risk of prostate cancer progression and mortality, but the molecular underpinnings of this association are poorly understood. Here, we demonstrate in a murine prostate cancer model, that high-fat diet (HFD) enhances the MYC transcriptional program through metabolic alterations that favour histone H4K20 hypomethylation at the promoter regions of MYC regulated genes, leading to increased cellular proliferation and tumour burden. Saturated fat intake (SFI) is also associated with an enhanced MYC transcriptional signature in prostate cancer patients. The SFI-induced MYC signature independently predicts prostate cancer progression and death. Finally, switching from a high-fat to a low-fat diet, attenuates the MYC transcriptional program in mice. Our findings suggest that in primary prostate cancer, dietary SFI contributes to tumour progression by mimicking *MYC* over expression, setting the stage for therapeutic approaches involving changes to the diet.

---

#A full list of authors and their affiliations appears at the end of the paper.

Prostate cancer is a leading cause of cancer-related lethality[1]. Systemic metabolic alterations can severely affect the course of the disease. Indeed, epidemiological studies have reported that saturated fat intake and obesity are associated with increased prostate cancer progression and mortality[2–5]. Considering the pandemic of obesity and diet-associated metabolic diseases[6–8] combined with the high frequency of newly diagnosed prostate cancers in developed countries, a better understanding of the mechanistic underpinnings of this link is of significant importance.

Preclinical and clinical studies have shown that systemic metabolic alterations associated with fat-enriched diets and obesity cooperate with tumour initiating genetic alterations to foster disease progression. Modulation of insulin/insulin-like growth factor 1 levels, phosphatidylinositol-3-kinase/mammalian target of rapamycin complex 1 pathway activation and pro-inflammatory stimuli have been implicated[9–14]. However, it is now clear that metabolic rewiring is tightly connected to changes at the epigenetic level as metabolites act as substrates or cofactors for epigenetic remodelling[15,16].

In prostate cancer, the landscape of epigenetic alterations varies greatly as the disease progresses from a confined tumour to the incurable castration-resistant metastatic stage[15,17]. However, the influence of metabolic alterations triggered by increased fat intake and/or obesity on prostate cancer epigenome rewiring and disease progression is still unexplored.

The oncogene c-MYC (MYC) is a key driver of human prostate cancer tumorigenesis and progression. MYC protein is over-expressed at early stages of the disease[18], whereas chromosome 8q gain, or focal amplification of 8q24.21, are associated with amplification of the MYC oncogene in primary prostate cancer, a feature exacerbated in metastatic disease and associated with poor disease-specific survival[19,20]. In the murine prostate, MYC over expression faithfully recapitulates the primary human disease[21].

A hallmark of MYC over expression in tumours is the induction of a global metabolic reprogramming to support cancer cell survival and growth[22–25]. Previous studies have shown that increased dietary fat intake significantly alters the biological behaviours of prostate cancers driven by MYC[10,11] suggesting this preclinical model as ideal to investigate the interplay between HFD, oncogene-driven metabolic vulnerabilities, and epigenetic alterations in prostate cancer progression.

Here, we integrate metabolome, epigenome and transcriptome profiling to identify HFD-driven alterations that foster prostate cancer progression in vivo. We demonstrate that increased fat intake amplifies MYC hallmarks and further enhances MYC's transcriptional program. Importantly, we identified a fat-induced MYC signature with clinical utility in identifying patients at higher risk of a more aggressive, lethal disease. Altogether, our findings suggest that a substantial subset of prostate cancer patients, including some without MYC amplification, may benefit from epigenetic therapies targeting MYC transcriptional activity or from dietary interventions targeting the metabolic dependencies regulated by MYC.

## Results

**HFD reprograms cancer metabolome and accelerates progression.** To examine the potential role of high-fat diet (HFD) in promoting metabolic rewiring of prostatic tissues, we compared mice that overexpress a human c-MYC transgene (MYC) in the prostate epithelium[21] to wild-type littermates (WT) that were fed either a HFD (60% kcal from fat; lard—rich in saturated fat) or a control diet (CTD; 10% kcal from fat; Supplementary Table 1). Irrespective of their genotype, mice that were fed with HFD developed the hallmarks of a diet-induced obesity phenotype, including increased body weight, liver steatosis, hyperinsulinemia, hyperglycaemia and a decrease in circulating 1,5-anhydroglucitol (a marker of short-term hyperglycaemia) (Fig. 1a and Supplementary Fig. 1a–e). At 12 weeks of age, MYC over expression, irrespectively of HFD, resulted in extensive cellular epithelium transformation to prostatic intraepithelial neoplasia (PIN) in the dorsolateral (DLP) and ventral (VP) prostate lobes, the latter with almost complete penetrance. Conversely, the anterior prostate (AP) remained mostly unaffected (Fig. 1b and Supplementary Fig. 1f). No presence of PIN was detected in the prostate lobes of WT animals fed a HFD (Supplementary Data 1). Increased tumour weight (Fig. 1c) and cell proliferation (Ki-67; Fig. 1d) were evident by 36 weeks of age in the HFD-fed mice compared to the CTD group, confirming previous reports that HFD significantly enhances the progression of MYC-driven prostate cancer[10,11].

The lack of a HFD-dependent phenotype at 12 weeks of age, combined with the robust and uniform transition to PIN triggered by MYC over expression observed in the VP (Fig. 1b, c and Supplementary Data 1), enabled us to investigate metabolic alterations driven by HFD before the appearance of a more aggressive, HFD-dependent phenotype. Untargeted metabolomics identified 414 metabolites in the prostate. As previously described[26], we confirmed that MYC induces a profound metabolic reprogramming in the VP affecting more than half of the metabolites detected, including metabolites related to glutamine, glucose, lipid, nucleotide metabolism and protein synthesis (Fig. 1e–g and Supplementary Data 2). Importantly, we found that these MYC-driven metabolic vulnerabilities were enhanced by HFD. Indeed, HFD resulted in increased levels of metabolites from glycolysis (i.e. lactate), glutaminolysis (i.e. glutamate), glutamine-metabolism related pathways including substrates, intermediates and final products of the citric acid cycle, nucleotide synthesis, amino acid metabolism (e.g. arginine, proline, aspartate and histidine), urea cycle, lipid metabolism and hexosamine biosynthesis (Fig. 1g and Supplementary Data 3); those features were also supported by Metabolite Set Enrichment Analysis (MSEA; Fig. 1h and Supplementary Data 4). Conversely, HFD had little impact on the WT prostatic metabolome, affecting only a total of 12 metabolites, nine of which were glycerophospholipids, and lowering 1,5-anhydroglucitol levels, in line with HFD-driven increase in circulating glucose and reduction of serum 1,5-anhydroglucitol (Fig. 1g, Supplementary Fig. 1d, e, g and Supplementary Data 5).

Notably, MYC over expression led to a significant decrease in s-adenosylmethionine (SAM), a member of the methionine cycle and the ultimate methyl donor required for methylation reactions (Fig. 1i). The donation of a methyl group by SAM results in its conversion to s-adenosylhomocysteine (SAH), which if accumulated, is a potent inhibitor of methyltransferases[27]. MYC also enhanced the levels of alpha-ketoglutarate (αKG), a critical co-factor for histone demethylation mediated by Jumonji Domain-containing Histone Demethylases (JHDM)[28]. Thus, these results suggest that histone methylation processes may be severely hindered during MYC-driven prostate cancer progression. Again, this feature was further exacerbated by diet since increased SAH levels (higher SAH/SAM ratio) were observed in the VP of HFD-fed mice (Fig. 1i and Supplementary Fig. 1h–i). Altogether, our data support the notion that HFD amplifies MYC-driven metabolic reprogramming.

**HFD enhances transcriptional changes at H4K20me1 dynamic genes.** To validate whether MYC/HFD affects histone methylation, we characterised 69 distinct combinations of histone modifications that span H2, H3, and H4 from all four genotype/diet

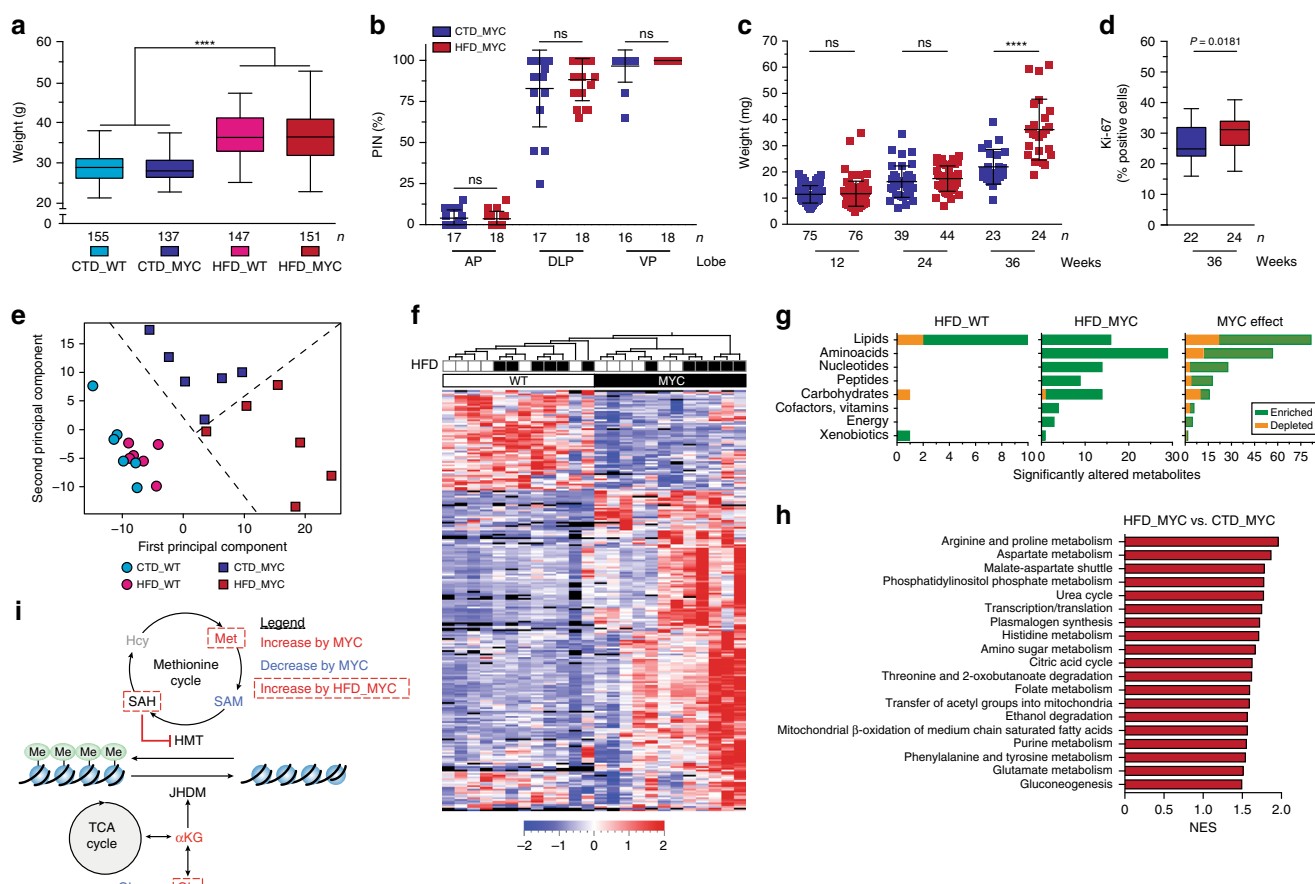

**Fig. 1** High-fat diet reprograms prostate cancer metabolome and accelerates disease progression. **a** Mice fed a high-fat diet (HFD) develop diet-induced obesity at 12 weeks of age ($n$ = biologically independent animals; two-way ANOVA, median, whiskers ± min/max; ****$P$ < 0.0001). **b**–**d** HFD does not alter the penetrance of prostatic intraepithelial neoplasia (PIN) at 12 weeks of age (**b**, $n$ = biologically independent lobes; unpaired $t$ test, mean ± s.d.; CTD: control diet; AP: anterior prostate; DLP: dorsolateral prostate; VP: ventral prostate; ns: non significant), but does lead to a greater tumour burden (**c**, $n$ = biologically independent lobes; Welch's $t$ test, mean ± s.d.; ****$P$ < 0.0001) and to cell proliferation, as assessed by Ki-67 (**d**, $n$ = biologically independent lobes; unpaired $t$ test, median, whiskers ± min/max) in the VP, at 36 weeks of age. **e** Principal component analysis identifies a distinct metabolic profile in the VP that is triggered by HFD, in a MYC context ($n$ = 6 biologically independent VP/condition, 414 metabolites detected). **f**, **g** Representation of all metabolites significantly altered by HFD in a WT ($n$ = 12) or a MYC ($n$ = 89) context, or by MYC overexpression irrespective of the diet ($n$ = 214) (**f**, unsupervised hierarchical clustering, $P$ < 0.05 and FDR < 0.15); the breakdown of metabolite classes is shown **g**. **h** Metabolite Set Enrichment Analysis (MSEA) revealed metabolic pathways significantly enriched by HFD in MYC-transformed VP ($P$ < 0.05 and FDR < 0.15). **i** Metabolic rewiring triggered by MYC and by HFD in a MYC context suggests dampened histone methylation. Hcy: homocysteine (undetected); Met: methionine; TCA: tricarboxylic acid (citric acid cycle); Gln: glutamine; Glu: glutamate. Source data are provided as a Source Data file

combinations in all murine prostatic lobes (DLP, VP, AP; Supplementary Data 6) by using a targeted mass spectrometry approach[29]. Unsupervised clustering of the different combinations of histone modifications revealed a strong MYC-driven signature in both DLP and VP (Fig. 2a). This was absent in the AP (Supplementary Fig. 2a) in line with the marginal PIN penetrance observed in this lobe (Fig. 1b). Among the histone peptides monitored, H3K27/K36 and H4K20 were significantly affected by MYC over expression. As previously described[30], MYC over expression induced a steep decrease in H3K27me3 (corresponding to the H3K27me3K36meX peptides). In particular, the H3K27me3 mark was hypomethylated in a stepwise process that can be catalysed by multiple JHDM enzymes and culminates with the unmethylated/acetylated H3K27 mark (Supplementary Fig. 2b). A similar pattern was observed for the H4K20 mark, but in this case the effect of MYC was significantly enhanced by HFD, leading to greater levels of the unmethylated mark (Fig. 2b). Importantly, HFD had no effect on the H4K20 mark in the WT tissues (Fig. 2b, c). H4K20me0 can be generated

from H4K20me1, a mark that is associated with transcriptional elongation[31], by the JHDM enzyme PHF8[32]. Chromatin immunoprecipitation followed by sequencing (ChIP-seq) of H4K20me1 revealed highly dynamic levels of this mark along each gene body upon MYC over expression with respect to the corresponding CTD_WT reference (Fig. 2d). Interestingly, modulation of the H4K20me1 mark at the gene body dictates levels of gene expression: thus, loss of H4K20me1 is associated with a decrease, while gain of H4K20me1 is associated with an increase in gene expression (Supplementary Fig. 2c). When comparing the gene expression levels for shared H4K20me1 dynamic gene body-associated regions between CTD_MYC and HFD_MYC conditions (Supplementary Fig. 2d), we found that the MYC-effect was systematically enhanced by HFD (Fig. 2e). These results suggest that HFD further enhances MYC-driven H4K20 hypomethylation leading to transcriptional changes.

**High-fat diet enhances MYC transcriptional activity.** To determine the cellular program specifically enhanced by HFD

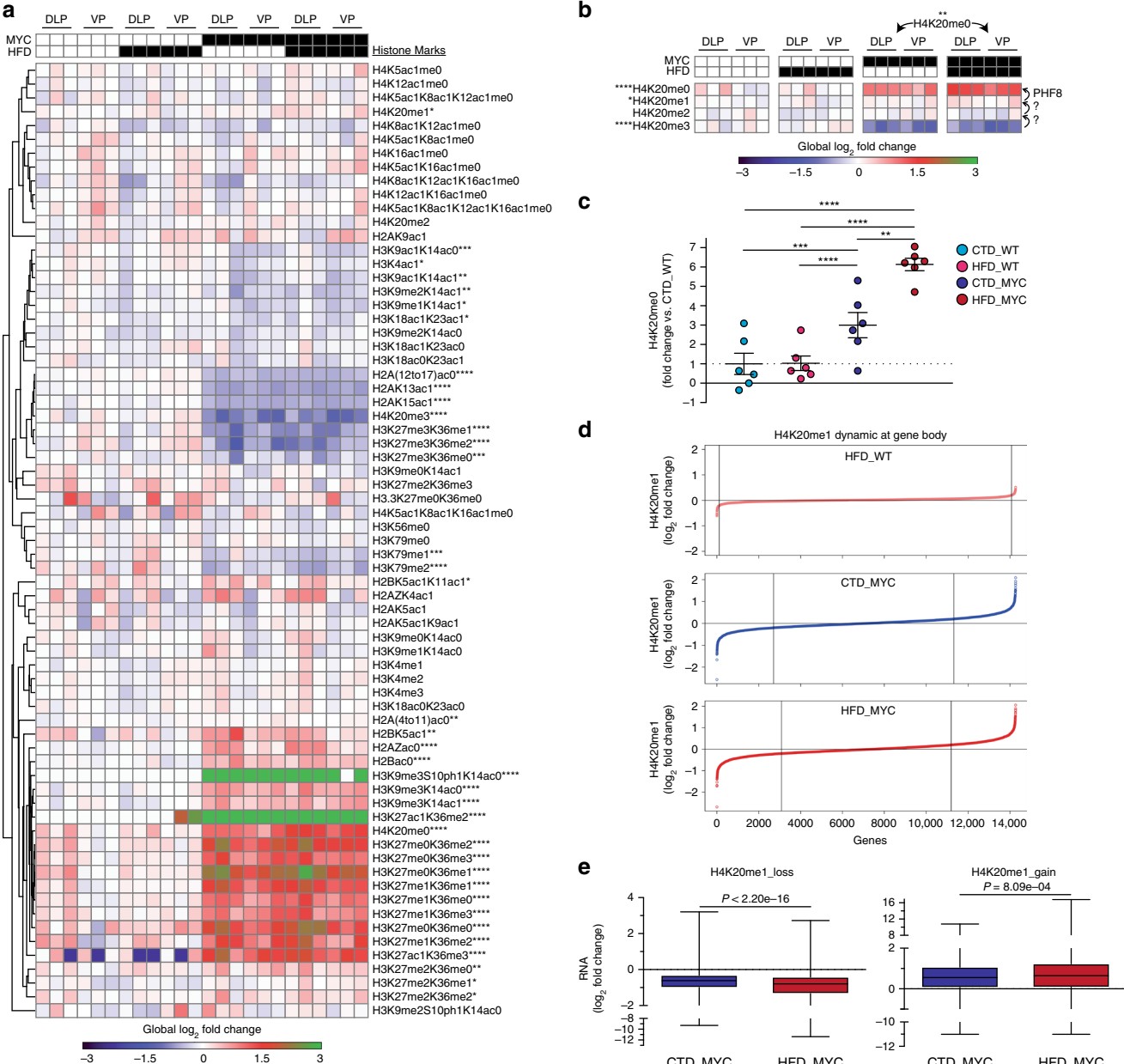

**Fig. 2** High-fat diet enhances MYC-driven transcriptional changes at H4K20me1 dynamic genes. **a** Global chromatin profiling identifies distinct chromatin-signature profiles in MYC-overexpressing DLP and VP lobes (histone marks levels relative to the DLP, VP and AP CTD_WT median values; MYC vs. WT comparisons, unpaired $t$ test; *$P < 0.05$, **$P < 0.01$, ***$P < 0.001$, ****$P < 0.0001$; Supplementary Data 6). **b** HFD enhances H4K20 hypomethylation triggered by MYC (MYC vs. WT and HFD_MYC vs. CTD_MYC comparisons, unpaired $t$ test; *$P < 0.05$, **$P < 0.01$, ****$P < 0.0001$). **c** H4K20me0 levels relative to the CTD_WT condition (fold change; unpaired $t$ test, mean ± SEM; **$P < 0.01$, ***$P < 0.001$, ****$P < 0.0001$). **d** H4K20me1 dynamic regions across all murine gene bodies relative to the CTD_WT condition. **e** HFD enhances the effect of MYC on gene expression at H4K20me1 dynamic regions that were lost (top, $n = 2508$ genes) or gained (bottom, $n = 3208$ genes; paired $t$ test, median, whiskers ± min/max)

within a MYC context, we performed Gene Sets Enrichment Analyses (GSEA) using the Hallmark gene sets (Supplementary Data 7)[33]. As expected, MYC over expression led to the enrichment of gene sets related to cell proliferation (E2F_targets, G2M_checkpoint), as well as MYC-transcriptional activity per se (Fig. 3a, left). Interestingly, HFD further enriched both gene sets related to MYC transcriptional activity (V1/V2), but only in MYC-transformed prostates (Fig. 3a, right). This feature was not linked to an increased expression of the MYC transgene (Supplementary Fig. 3a). Because the MYC transcriptional program is highly context-specific[34], we generated a murine prostatic MYC

signature by including the leading edge genes ($n = 610$) of MYC-related gene sets that were significantly enriched by MYC and/or HFD feeding (Supplementary Fig. 3b and Supplementary Data 8–9). As expected, the expression levels of MYC signature genes were elevated following MYC over expression, and further increased by HFD (Fig. 3b). ChIP-seq of PHF8, the JHDM that mediates H4K20me1 demethylation[32] and a known MYC transcriptional coactivator and regulator of proliferation[35,36], revealed that MYC over expression increases the recruitment of PHF8 to the promoter regions of MYC signature genes. Again, we observed that this effect was enhanced by HFD (Fig. 3c).

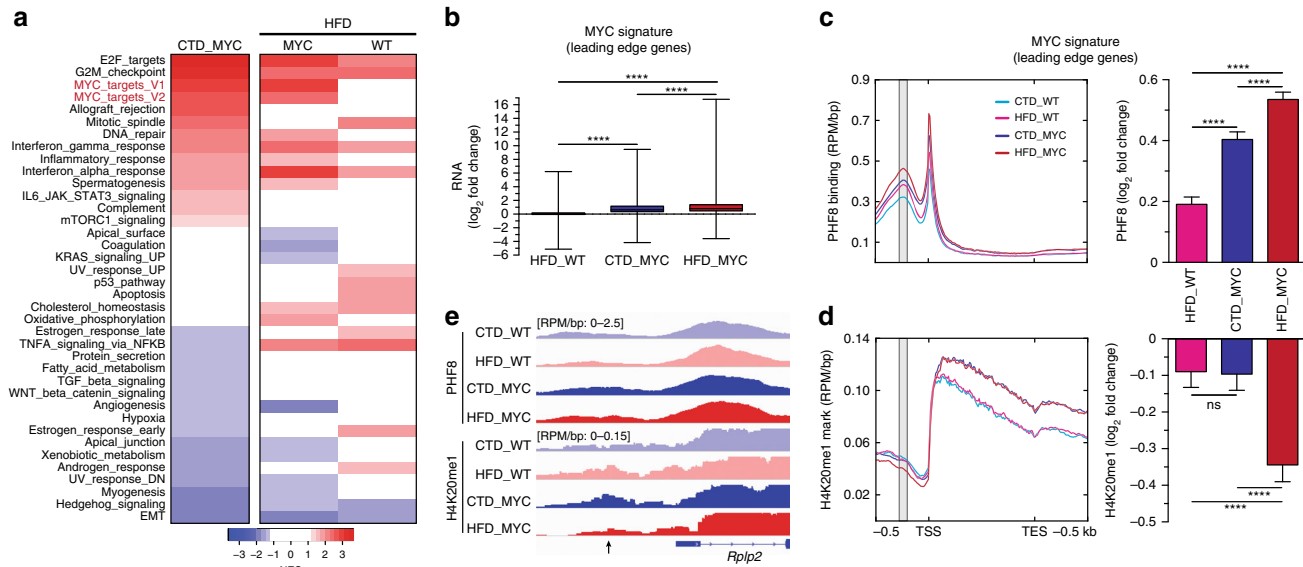

**Fig. 3** High-fat diet enhances MYC transcriptional activity. **a** Gene Set Enrichment Analysis (GSEA, Hallmark, $P < 0.05$ and FDR < 0.1) revealed enhanced expression of MYC target genes triggered by HFD in the MYC context (right column-left side: HFD_MYC vs. CTD_MYC; right column-right side: HFD_WT vs. CTD_WT comparisons; left column: CTD_MYC vs. CTD_WT comparison). **b** HFD boosts the expression of the murine prostatic-derived MYC signature ($n = 610$ genes; fold-change relative to the CTD_WT condition, paired $t$ test, median, whiskers ± min/max; ****$P < 0.0001$). **c** Increased PHF8 recruitment at the promoter (grey; −270 to −210 bp) of MYC signature genes, is triggered by MYC overexpression and boosted by HFD (fold-change relative to the CTD_WT condition, paired $t$ test, mean ± SEM; ****$P < 0.0001$). **d** Depletion in H4K20me1 mark at the promoter of MYC signature genes (grey) requires increased PHF8 recruitment together with HFD (paired $t$ test, mean ± SEM; ****$P < 0.0001$). **e** Representative PHF8 and H4K20me1 tracks at Rplp2 promoter (arrow), a MYC signature leading edge gene (RPM/bp: reads per million/base pair)

However, only when MYC over expression was combined with HFD, was a significant decrease in H4K20me1 observed at PHF8 recruitment sites (Fig. 3d, e). Taken together, these results suggest that the observed HFD-induced enhancement of MYC transcriptional program is, at least in part, mediated via an increased recruitment and activity of PHF8 toward the H4K20me1 mark at MYC signature genes. This program then culminates in augmented cell proliferation and tumour burden (Fig. 1c, d).

**A SFI MYC signature is associated with lethal disease.** Since our results in the preclinical model represent the combined effects of both increased dietary animal fat intake (AFI) and the diet-induced obesity phenotype, we next investigated whether dietary AFI, independently of obesity, could recapitulate the HFD-driven phenotype in humans. We used data on AFI, as documented in the Health Professionals Follow-up Study (HPFS) and Physicians' Health Study (PHS) cohorts, to stratify the 319 prostate cancer patients for whom tumour (genetic background uncharacterised) and adjacent-normal gene expression profiles were available (Table 1). Using GSEA analysis, we identified the MYC_targets_V1 among the three gene sets that were significantly enriched by high AFI, while no gene set was enriched in the adjacent-normal prostatic tissues (Fig. 4a and Supplementary Data 10). When the leading edge genes within the AFI-induced MYC_targets_V1 signature (Fig. 4b, Table 1 and Supplementary Data 11) were used to create a metagene score, we found that prostate cancer patients with greater AFI-dependent MYC transcriptional activation in their tumour tissues were more likely to die of their disease ($n$ lethal = 31, Odds Ratio (OR) = 3.44, 95% CI = 1.69–7.38). This holds true after adjusting for gleason grade and body mass index (BMI; Table 2). Conversely, when we used the MYC signature genes not enriched by AFI (non-leading edges genes) to create a similar metagene score, this score was unable to predict lethal disease after adjusting for gleason grade and BMI

(Table 2). We next investigated which type of fatty acid contributes to the enrichment of the MYC transcriptional program. We identified the MYC_targets_V1 as the top gene set that was enriched by high saturated fat intake (SFI; Fig. 4a, and Supplementary Data 12), while neither monounsaturated nor polyunsaturated fat intake was associated with a positive enrichment of any given gene set (Fig. 4a and Supplementary Data 13–14). Importantly, the SFI-induced MYC_targets_V1 signature was more robustly associated with prostate cancer lethality ($n$ lethal = 34, Odds Ratio (OR) = 4.02, 95% CI = 1.98–8.63; Tables 1 and 2), a feature that was not recapitulated when using a randomly picked MYC_targets_V1 signature (Supplementary Table 2). Furthermore, the metagene score was more strongly related to lethal disease in men with a high SFI than in men with a low SFI ($P$ for interaction = 0.03; Fig. 4c). These results indicate that the MYC-transcriptional program specifically induced by SFI drives prostate cancer lethality.

We confirmed the prognostic value of the SFI-induced MYC signature in four independent clinical cohorts by analysing gene expression in the tumours from 631 prostate cancer patients. Strikingly, even in these cohorts lacking patient dietary information, the high expression of the SFI-induced MYC signature identified patients that were more likely to progress to a metastatic disease in the Thomas Jefferson University (TJU), Johns Hopkins Medical Institutions-I (JHMI-I), Mayo Clinic and Cedar-Sinai cohorts ($P = 1.33e-04$), a feature that was much less pronounced when using the non-SFI-associated MYC signature ($P = 1.26e-02$; Fig. 4d). Importantly, in patients from the TJU/JHMI-I/II cohorts, the SFI-induced MYC signature was not associated with BMI (Supplementary Fig. 4). Additional univariate and multivariate analyses confirmed the prognostic power of the SFI-induced MYC signature in predicting prostate cancer progression to a metastatic disease, even after adjusting for gleason grade or the Cell Cycle Progression score consisting of 31 cell cycle genes (Supplementary Tables 3–5)[37]. Altogether, these

**Table 1 Characteristics of 319 men diagnosed with prostate cancer from 1982 to 2005 in the Health Professionals Follow-up Study and the Physicians' Health Study according to fat intake MYC metagene scores**

| Characteristic | All men (n = 319) | Animal fat MYC metagene score[a] | | | Saturated fat MYC metagene score[a] | | |
|---|---|---|---|---|---|---|---|
| | | Tertile 1 (low) (n = 107) | Tertile 2 (n = 106) | Tertile 3 (high) (n = 106) | Tertile 1 (low) (n = 106) | Tertile 2 (n = 107) | Tertile 3 (high) (n = 106) |
| Age at diagnosis, years, mean (SD) | 65.0 (6.3) | 65.6 (6.2) | 64.9 (6.6) | 64.5 (6.1) | 65.6 (6.2) | 64.7 (6.3) | 64.7 (6.3) |
| Year of diagnosis, n (%) | | | | | | | |
| Before 1990 (pre-PSA era) | 28 (8.8) | 9 (8.4) | 10 (9.4) | 9 (8.5) | 8 (7.4) | 12 (11.3) | 8 (7.5) |
| 1990–1993 (peri-PSA era) | 83 (26.0) | 32 (29.9) | 27 (25.5) | 24 (22.6) | 32 (29.9) | 27 (25.5) | 24 (22.6) |
| After 1993 (PSA era) | 208 (65.2) | 66 (61.7) | 69 (65.1) | 73 (68.9) | 67 (62.6) | 67 (63.2) | 74 (69.8) |
| BMI at diagnosis, kg/m², mean (SD) | 25.2 (2.9) | 24.7 (2.6) | 25.4 (2.9) | 25.5 (3.1) | 24.7 (2.6) | 25.4 (2.8) | 25.6 (3.2) |
| PSA at diagnosis, ng/ml, median [25th – 75th percentile][b] | 7.4 [5.3–11.6] | 7.0 [5.0, 11.9] | 7.6 [5.5, 12.9] | 8.0 [5.6, 11.1] | 6.9 [5.0-12.0] | 7.6 [5.4-13.0] | 7.9 [5.6-11.1] |
| Pathologic TNM stage, n (%)[c] | | | | | | | |
| T2 N0 M0 | 192 (61.9) | 73 (69.5) | 60 (58.3) | 59 (57.8) | 72 (68.6) | 60 (58.3) | 60 (58.8) |
| T3 N0 M0 | 107 (34.5) | 29 (27.6) | 40 (38.8) | 38 (37.3) | 30 (28.6) | 40 (38.8) | 37 (36.3) |
| T4/N1/M1 | 11 (3.5) | 3 (2.9) | 3 (2.9) | 5 (4.9) | 3 (2.9) | 3 (2.9) | 5 (4.9) |
| Clinical TNM stage, n (%)[d] | | | | | | | |
| T1/T2 N0 M0 | 297 (93.4) | 103 (96.3) | 98 (92.5) | 96 (91.4) | 102 (95.3) | 99 (93.4) | 96 (91.4) |
| T3 N0 M0 | 21 (6.6) | 4 (3.7) | 8 (7.5) | 9 (8.6) | 5 (4.7) | 7 (6.6) | 9 (8.6) |
| Gleason grade, n (%) | | | | | | | |
| <7 | 51 (16.0) | 24 (22.4) | 16 (15.1) | 11 (10.4) | 22 (20.6) | 17 (16.0) | 12 (11.3) |
| 3+4 | 124 (38.9) | 49 (45.8) | 39 (36.8) | 36 (34.0) | 49 (45.8) | 40 (37.7) | 35 (33.0) |
| 4+3 | 81 (25.4) | 19 (17.8) | 32 (30.2) | 30 (28.3) | 20 (18.7) | 32 (30.2) | 29 (27.4) |
| >7 | 63 (19.7) | 15 (14.0) | 19 (17.9) | 29 (27.4) | 16 (15.0) | 17 (16.0) | 30 (28.3) |
| Tissue type, n (%) | | | | | | | |
| RP | 311 (97.4) | 105 (98.1) | 103 (97.2) | 103 (97.2) | 105 (98.1) | 103 (97.2) | 103 (97.2) |
| TURP | 8 (2.5) | 2 (1.9) | 3 (2.8) | 3 (2.8) | 2 (1.9) | 3 (2.8) | 3 (2.8) |
| Cohort, n (%) | | | | | | | |
| HPFS | 213 (66.8) | 61 (57.0) | 73 (68.9) | 79 (74.5) | 65 (60.7) | 68 (64.2) | 80 (75.5) |
| PHS | 106 (33.2) | 46 (43.0) | 33 (31.1) | 27 (25.5) | 42 (39.3) | 38 (35.8) | 26 (24.5) |

*SD* standard deviation, *PSA* prostate-specific antigen, *BMI* body mass index, *TNM* tumour, lymph node, metastasis, *RP* radical prostatectomy, *TURP* transurethral resection of the prostate, *HPFS* Health Professionals Follow-up Study, *PHS* Physicians' Health Study
[a]The genes identified in the enrichment analysis of MYC_targets_V1 pathway in tumour tissues were used to create a metagene score. A score was computed for each sample by averaging the normalised (mean-centered and variance scaled) expression values of all member genes. The score was divided into tertiles
[b]29 men missing PSA at diagnosis
[c]Nine men missing pathologic TNM stage
[d]One man missing clinical TNM stage

results demonstrate that high SFI, independent of obesity or features of it, fosters a MYC-driven cellular program, promoting the progression to a metastatic and lethal disease.

Finally, we investigated whether a dietary intervention could reverse the HFD-induced MYC transcriptional program. While the HFD robustly enhanced the MYC transcriptional program induced by MYC over expression in the murine prostate, switching to a CTD at 10 weeks of age was sufficient to dampen the MYC_targets_V1 signature observed in 12-week-old mice (Fig. 4e). This suggests that a dietary intervention aimed at lowering AFI and potentially more importantly SFI in patients might be able to directly impact the MYC transcriptional program, thereby reducing or delaying the progression to a lethal, metastatic disease.

## Discussion

In this study, we report the effect of HFD-mediated systemic alterations on prostate cancer progression. Our data demonstrate that HFD synergises with oncogenic transformation of the prostate to promote a MYC-driven program and disease progression. In the normal prostate, HFD impacts metabolites that are primarily restricted to membrane lipid remodelling, has little influence on histone modifications, and results in a distinct transcriptional program compared to that induced by HFD in the transformed prostate. Conversely, HFD profoundly alters an early

stage of MYC-induced prostate transformation characterised by PIN, resulting in the enhancement of MYC-driven metabolic, epigenetic, and transcriptional programs (Fig. 4f). These data suggest that a premalignant condition such as PIN, which often precedes the onset of invasive adenocarcinoma in humans[38], is required for HFD to exert its MYC-amplifying effects in the prostate.

A substantial body of literature supports the notion that cellular metabolism has a profound influence on epigenetic modifications, which rely on metabolites as substrates or cofactors[39–42]. Here, we provide the evidence that HFD acts as a master effector of prostate cancer metabolism, creating an environment that favours histone hypomethylation and results in an enhanced MYC-driven transcriptional program. Notably, we observed a decrease in the H4K20me1 mark at the promoter region of MYC signature genes, a feature that was associated with both an increased recruitment and activity of PHF8, a JHDM and the only enzyme known to demethylate the H4K20me1 mark[32]. Along this line, PHF8 has been documented in cell culture systems as a MYC transcriptional coactivator[35] and a regulator of prostate cancer cell proliferation, migration and invasion[36,43], supporting the idea that an incremental gain in PHF8 activity by HFD at H4K20me1 might provide increased tumour fitness over the course of prostate cancer development.

The unifying aspect that translates from mice to humans is the fat-induced MYC signature, a feature that persists despite the

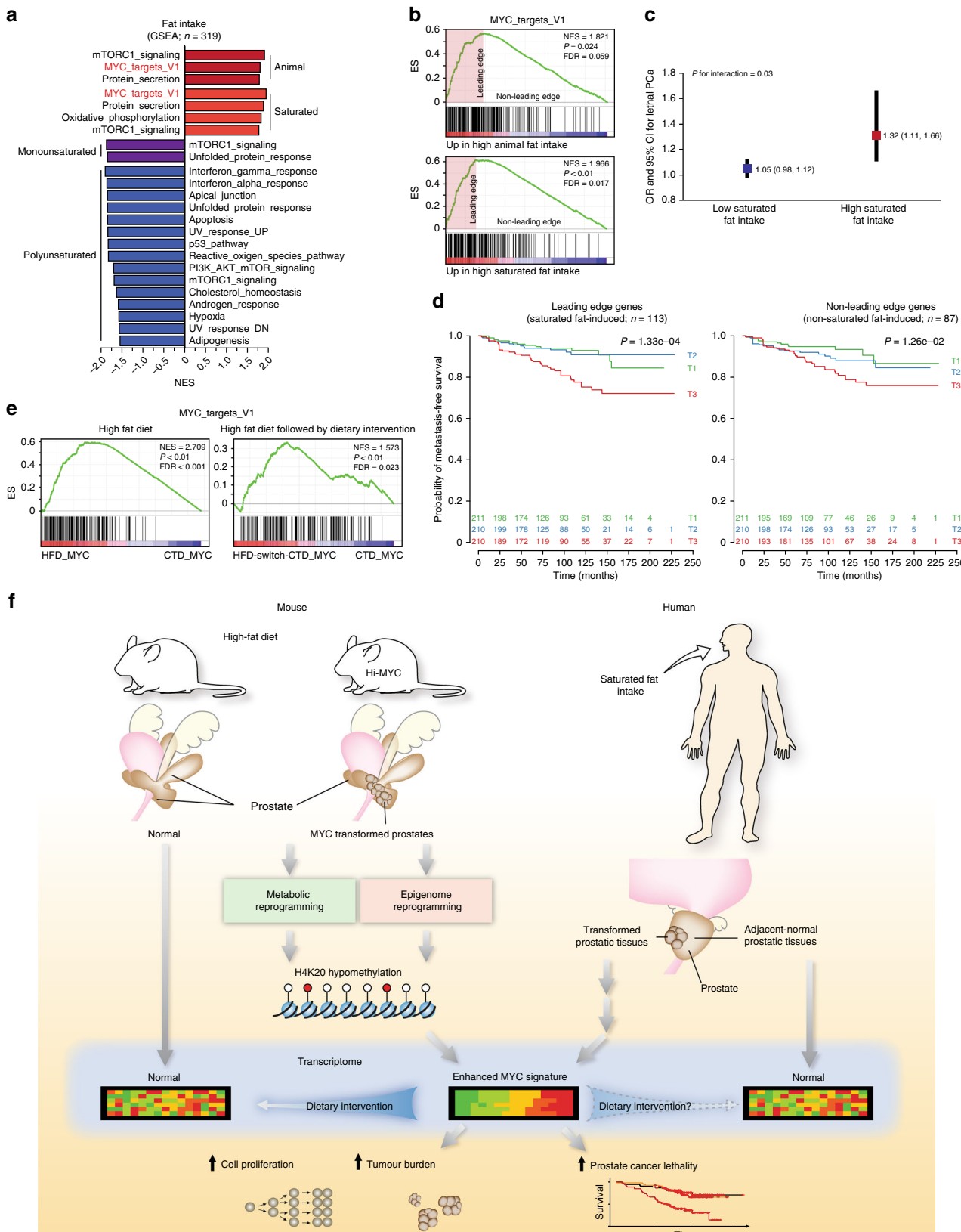

**Fig. 4** A saturated fat-induced MYC signature is associated with lethal prostate cancer. **a**, **b** GSEA analysis (Hallmark) revealed that high animal fat and high saturated fat intake enriches for the MYC_targets_V1 gene set (**a**, $P < 0.05$ and FDR < 0.1), as represented by the enrichment plot (**b**) in the HSPH/PHS cohorts. **c** The lethality for every 0.1 unit increase of MYC score was significantly elevated among patients with high saturated fat intake compared with those with low saturated fat intake. **d** High expression of the saturated fat-induced MYC signature is significantly associated with reduced metastatic-free survival (T3) in four independent cohorts (TJU/JHMI-I/Mayo Clinic/Cedar-Sinai cohorts, $n = 631$). **e** Short-term dietary intervention (HFD switch to CTD) dampens the HFD-induced MYC transcriptional activity in MYC-driven murine prostate cancer. **f** Graphical summary

**Table 2 Fat-induced and non-fat-induced MYC signature score in relation to risk of prostate cancer death among men diagnosed with non-metastatic prostate cancer**

| MYC score | n | Leading edge genes (fat-induced)[a] | | | n | Non-leading edge genes (non-fat-induced)[b] | | |
|---|---|---|---|---|---|---|---|---|
| | | OR (95% CI)[c] | OR (95% CI)[d] | OR (95% CI)[e] | | OR (95% CI)[c] | OR (95% CI)[d] | OR (95% CI)[e] |
| Animal fat | | | | | | | | |
| Tertile 1 (low) | 13 | 1.00 | 1.00 | 1.00 | 17 | 1.00 | 1.00 | 1.00 |
| Tertile 2 | 18 | 1.58 (0.73, 3.53) | 1.31 (0.57, 3.08) | 1.27 (0.55, 2.99) | 19 | 1.17 (0.57, 2.44) | 1.03 (0.47, 2.30) | 0.96 (0.43, 2.16) |
| Tertile 3 (high) | 31 | 3.44 (1.69, 7.38) | 2.50 (1.14, 5.70) | 2.37 (1.07, 5.43) | 26 | 1.79 (0.90, 3.64) | 1.07 (0.81, 3.70) | 1.66 (0.78, 3.61) |
| *P*, linear trend[f] | | 0.001 | 0.019 | 0.03 | | 0.09 | 0.15 | 0.17 |
| Saturated fat | | | | | | | | |
| Tertile 1 (low) | 13 | 1.00 | 1.00 | 1.00 | 16 | 1.00 | 1.00 | 1.00 |
| Tertile 2 | 15 | 1.23 (0.55, 2.80) | 1.07 (0.45, 2.59) | 1.05 (0.44, 2.54) | 18 | 1.24 (0.59, 2.64) | 1.17 (0.52, 2.65) | 1.09 (0.48, 2.48) |
| Tertile 3 (high) | 34 | 4.02 (1.98, 8.63) | 3.21 (1.47, 7.35) | 3.04 (1.38, 7.01) | 28 | 2.34 (1.18, 4.82) | 1.93 (0.90, 4.23) | 1.86 (0.87, 4.08) |
| *P*, linear trend[f] | | 0.0001 | 0.002 | 0.004 | | 0.015 | 0.085 | 0.107 |

*n* lethal events, *OR* odds ratio; *CI* confidence interval
[a]Animal fat: *n* = 122; Saturated fat: *n* = 113
[b]Animal fat: *n* = 78; Saturated fat: *n* = 87
[c]Logistic regression model adjusted for age and year at diagnosis (continuous)
[d]Logistic regression model adjusted for age and year at diagnosis (continuous), and gleason grade (continuous: <7, 3 + 4, 4 + 3 >7)
[e]Logistic regression model adjusted for age and year at diagnosis (continuous), gleason grade (continuous: <7, 3 + 4, 4 + 3, >7), and BMI at diagnosis (continuous)
[f]Estimated by modeling tertiles of MYC score as continuous variable (tertile 1 = 0, tertile 2 = 1, tertile 3 = 2)

genetic heterogeneity of human prostate cancers[19,44]. Indeed, dietary fat intake does not only amplify the MYC transcriptional program in MYC-driven prostate cancers, but can enrich for it, even in cancers lacking MYC over expression. This suggests that the enhancement of MYC-driven metabolic and epigenetic reprogramming may be a general mechanism that underlies the influence of dietary fat intake on prostate cancer progression although this hypothesis remains to be tested across prostate cancer molecular subtypes.

Finally, we show that the saturated fat-induced MYC transcriptional signature is not only a tumour biomarker for the patients' diet, but it is prognostic for progression to metastatic, lethal prostate cancer. Indeed, the SFI-induced MYC signature is able to predict prostate cancer lethality, independently of the degree of tumour differentiation and patient obesity status and the robustness of this finding was validated in four independent cohorts. Importantly, the non-fat-induced MYC signature was unable to predict lethal disease in the HPFS/PHS cohorts and was only marginally significant as prognostic marker in the validation cohorts. This highlights the fact that saturated fat intake not only enriches the expression of MYC-regulated genes but does so especially for the most predictive subset of genes, possibly reflecting the convergence of oncogenic signalling pathways on dysregulated lipid metabolism, a key feature for prostate cancer development and progression to a metastatic disease[45].

Several in vitro studies showed that MYC drives the de novo synthesis of fatty acids and increases the expression of the key lipogenic enzymes such as the ATP Citrate Lyase (ACLY), Acetyl-CoA carboxylase (ACC) and fatty acid synthase (FASN)[46–48]. Moreover, previous metabolic profiling performed by our group on MYC engineered prostate cells, in vivo models, and human prostate cancers showed that MYC overexpression specifically enhances lipid metabolism underlying lipid deregulation as key metabolic feature of MYC oncogenic activity in prostate cancer[26]. Recently HFD has been shown to induce an intra-tumoural lipogenic signature driving metastasis formation in *Pten* deficient mouse model[49]. We are currently performing lipidomics analysis to investigate the interplay between diet-derived fats and MYC-driven de novo lipogenesis in shaping the tumour lipidome and promoting a more aggressive phenotype.

Collectively, our results suggest that extrinsic risk factors—such as saturated fat intake—contribute to prostate cancer lethality by enriching for a MYC-transcriptional program; and either

synergise with *MYC* over expression, which is observed in 37% of metastatic prostate cancers[50], or phenocopy *MYC* amplification (Fig. 4f). While neither MYC protein nor MYC mRNA over-expression measured in primary tumours from patients subjected to radical prostatectomy are strong prognostic markers[51], our findings suggest that a SFI-dependent MYC signature can be used in the clinical setting to identify patients with a worse prognosis. Finally, our study raises the possibility that a nutritional program such as that involving the reduction of animal fat and specifically saturated fat consumption in men with early-stage cancer may dampen the MYC transcriptional program and diminish or delay the risk of disease progression.

## Methods

**Animal husbandry.** FVB Hi-MYC mice (strain number 01XK8), expressing the human c-MYC transgene in prostatic epithelium, were obtained from the National Cancer Institute Mouse Repository at Frederick National Laboratory for Cancer Research[21]. Upon weaning (3 weeks), male mice heterozygous for the transgene (MYC), together with their wild type littermates (WT), were fed a purified control diet (CTD; Harlan Laboratories, TD.130838) consisting of 10% fat, or a high-fat diet (HFD; Harlan Laboratories, TD. 06414) consisting of 60% fat (Supplementary Table 1) until 12, 24 or 36 weeks of age; ingredients were adjusted on a kcal basis (Supplementary Table 6). For dietary intervention experiments, mice assigned an HFD were switched to a CTD at 10 weeks of age for the folllowing 2 weeks until the experimental endpoint. Litters were randomly assigned to each diet. Group allocation was performed in a non-blinded fashion. Food was changed on a weekly basis, and mice were weighed every three weeks, starting at weaning. Animals were kept on a 12-h light/12-h dark cycle, and allowed free access to food and water at the Dana-Farber Cancer Institute (DFCI) Animal Resources Facility. The animal protocol was reviewed and approved by the DFCI Institutial Care and Use Committee (IACUC), and was in accordance with the Animal Welfare Act. Mice sample size estimate for analyses was based on published literature.

**Tissue collection.** At defined time points, mice were weighed and euthanized by $CO_2$, followed by cervical dislocation; blood was collected by cardiac puncture, and serum was collected using serum-separating tubes (#41.1378.005, Sarstedt), aliquoted, and stored at −80 °C. Urogenital apparatus and liver tissues were fixed in 10% buffered formalin and processed for paraffin embedding. Alternatively, mouse prostate lobes (anterior prostate, AP; dorsolateral prostate, DLP; ventral prostate, VP) were immediately dissected, weighed and flash-frozen in liquid nitrogen. Serum and tissues were consistently collected during the same periods to minimise inter-samples and circadian rhythm variability.

**Histopathologic and immunohistochemical analyses.** Formalin-fixed, paraffin-embedded mouse urogenital apparatus and liver tissues were sectioned (5 μm) and stained with hematoxylin and eosin (H&E). Histopathological slides were analysed by expert murine uropathologist, who were blind to the experimental conditions. Hepatic steatosis was also assessed for liver tissues (M.L.). The presence and extent

of PIN in 12-weeks-old mice (AP, DLP, VP) was estimated for each mouse, by evaluating the percentage of the gland affected for each prostate lobe and reported in Supplementary Data 1 (M.L.). For Ki-67 staining, slides were baked for 60 min in an oven set to 60 °C. They were then loaded into the Bond III staining platform with appropriate labels. Slides were antigen retrieved in Bond Epitope Retrieval 2 for 20 min, and incubated with rabbit monoclonal anti-Ki-67 antibody (#VP-RM04 (clone SP6), Vectors Laboratories) at dilution 1:250 for 30 min, room temperature. Primary antibody was detected using Bond Polymer Refine Detection kit. Slides were developed in 3,3′-diaminobenzidine (DAB), dehydrated, and coverslipped. The percentage of Ki-67 positive cells was evaluated by counting the number of cell that expresses nuclear Ki-67 as a function of the total number of cells per high power field. Whenever possible, up to 10 high power fields for each VP lobe were counted, averaged, and counts were reported as each sample's score (F.G. and M. F.). Sample size for histological evaluation was estimated based on previous literature data, using the same model[10]. For Ki-67 analysis, we performed sample size calculation using the software G*power version 3.1, extrapolating the effect size (d = around 0.87) from the data of Kobayashi et al.[11] in MYC mice fed with HFD. Based on this assumption, we calculated that at least 22 mice/group should be used to detect a significant difference in Ki-67 positivity using a two-sided t-test for change in mean between two independent groups, with an alpha-error of 0.05 and a priori power of 0.8.

**Insulin ELISA**. Serum insulin levels were measured using an insulin-1 ELISA kit from Sigma-Aldrich (#RAB0817). Briefly, samples were diluted 1:3 or 1:5 in diluent buffer C (provided in the kit) and the assay was performed according to the manufacturer's instructions. Each sample was measured twice (technical duplicate). Outliers (identified using the ROUT method, Q = 0.1%), and samples in which insulin levels were under the detection limit of the assay, were removed from the analysis. Statistical analysis and graphical representation were performed with use of GraphPad Prism version 7.0.

**Metabolic profiling**. For Metabolic profiling of serum and prostatic tissues (VP), we used the platform from Metabolon Inc. (Durham, NC, USA). Mice sample size to ensure adequate power for metabolomics analysis was based on previous literature data using a similar model[26]. Information regarding sample preparation, quality assurance (QA) and control (QC), and metabolite quantification was provided by the company as follows:

*Sample preparation*: Biological samples were stored at −80 °C and then thawed on ice just prior to extraction. Tissue samples were weighed at Metabolon on a 4-position analytical scale (1/10th mg) and then soaked overnight in 80% methanol/ 20% deionized water with recovery standards at a 60 μL: 1 mg ratio. The methanol contained four recovery standards (DL-2-fluorophenylglycine, tridecanoic acid, d6-cholesterol and 4-chlorophenylalanine) to allow confirmation of extraction efficiency. For serum, 100 μl sample volume was extracted with 500 μl of methanol containing recovery standards. All extracts were divided into four fractions: one for Ultra-performance liquid chromatography tandem mass-spectrometry (UPLC-MS/ MS) with positive ion mode electrospray ionisation (IMEI); one for (UPLC-MS/ MS) with negative IMEI; one for liquid chromatography (LC) polar platform; the final fraction was reserved as a backup. Aliquots were dried and then the first aliquot was reconstituted in 80 μL of 6.5 mM ammonium bicarbonate in water (pH 8) for the negative ion analysis, the second aliquot was reconstituted using 80 μL 0.1% formic acid in water (pH ~3.5) for the positive ion method, while the third aliquot was reconstituted in 80 μL of hydrophilic interaction liquid chromatography (HILIC) solvent (15% $H_2O$: 5% MeOH: 80% ACN) with 10 mM ammonium formate (pH~10) for the HILIC method.

*QA/QC*: Several types of controls were analysed together with the experimental samples: (1) a pooled matrix sample specific for each sample type (i.e. prostate and serum) was generated by combining 20 μl of each experimental sample and injecting the pooled sample six times for each data set to serve as a technical replicate to assess process variability; (2) five water aliquots were extracted and analysed to serve as process blanks for artifact determination; (3) a cocktail of internal standards, carefully chosen to not interfere with the measurement of endogenous compounds, was spiked into every analysed sample to monitor instrument performance and serve as retention markers for chromatographic alignment. The list of internal standards is provided in Supplementary Table 7. Instrument variability was evaluated during the entire procedure. Experimental samples were randomised across the platform run.

*UPLC Method*: Separations were performed using a Waters Acquity UPLC (Waters, Milford, MA). Reverse-phase (RP) positive ion method analysis used mobile phase consisting of 0.1% formic acid in water (A) and 0.1% formic acid in methanol (B). Reverse-phase negative ion analysis used mobile phase consisting of 6.5 mM ammonium bicarbonate in water, pH 8 (A) and 6.5 mM ammonium bicarbonate in 95% methanol/5% water (B). The sample injection volume was 5 μL and a 2x needle loop overfill was used. Separations utilised separate acid and base-dedicated 2.1 mm × 100 mm Waters BEH C18 1.7 μm columns held at 40 °C. HILIC used mobile phase consisting of 10 mM ammonium formate in 15% water, 5% methanol, 80% acetonitrile (effective pH 10.16 with NH4OH) (A) and 10 mM ammonium formate in 50% water, 50% acetonitrile (effective pH 10.60 with NH4OH) (B). The sample injection volume was identical to RP method. The stationary phase consisted of a 2.1 mm × 150 mm Waters BEH Amide 1.7 μm

column held at 40 °C. The gradient profiles for RP and HILIC methods can be found in Supplementary Table 8.

*High Resolution Accurate Mass (HRAM) method*: A ThermoFisher Scientific (Waltham, MA) Q-Exactive was the HRAM instrument used[52]. Detailed source and MS settings can be found in Supplementary Table 9 (conditions are also described in supplementary information from Evans et al.)[53]. The scan range was 80–1000 *m/z* with a scan speed of ~9 scans per second (alternating between MS and MS/MS scans), and the resolution was set to 35,000 (measured at 200 *m/z*). Mass calibration was performed as needed to maintain <5 ppm mass error for all standards monitored.

*Biological sample analysis*: Metabolon has developed a chemocentric approach that was used in peak detection and integration, and is described in detail elsewhere[54–56]. This in-house peak detection and integration software was used, the data output of which was a list of *m/z* ratios, retention indices (RI) and area under the curve (AUC) values. User specified criteria for peak detection included thresholds for signal to noise ratio, area and width. Relative standard deviations (RSDs) of peak area were determined for each internal and recovery standard to confirm extraction efficiency, instrument performance, column integrity, chromatography and mass calibration. The biological data sets, including QC samples, were chromatographically aligned based on a retention index that utilised internal standards assigned a fixed RI value. The RI of the experimental peak was determined by assuming a linear fit between flanking RI markers whose RI values are set. Peaks were matched against an in-house library of authentic standards and routinely detected unknown compounds specific to the respective method. The library consisted of 3200 endogenous and exogenous metabolites for which super and subpathway designations were provided. Identifications were based on retention index values, experimental precursor mass match to the library authentic standard within 10 ppm, and quality of MS/MS match. MS/MS forward and reverse match scores were based on a comparison of the ions present in the experimental spectrum to the ions present in the library spectrum. A forward score of 100 would mean all the ions present in the experimental spectrum were present in the library at the correct ratios. Any deviations in ion ratios or additional experimental ions not present in the library reduced the forward score, thus the forward score is a good indication of the purity of the compound being detected. Co-elution with another molecule with the same mass add ions to the experimental spectrum and reduce the forward score. Similarly, a reverse score of 100 indicated that all ions present in the library were present in the experimental spectrum at the correct ratios and deviations in ion ratios or ions in the library not present in the experimental spectrum reduced the reverse score. Identification was automatically approved if all the above criteria were met and the MS/MS forward and reverse scores were above 80. Compounds which met the above criteria but had low MS/ MS scores, below 35 for both forward and reverse, were automatically rejected. Compounds with intermediate MS/MS forward and reverse scores, 36–79, were marked for manual review. If an MS/MS spectrum was not obtained for a given ion, the identification was based on retention and parent mass alone and marked for analyst reviews. In this case, identification can still be confirmed if it has historical precedent in the specific matrix. Further details can be found in Evans et al.[56].

*Metabolite quantification and data normalisation*: Peaks were quantified using area-under-the-curve. Data was normalised, to correct variations that resulted from differences in the inter-day tuning of the instruments. Essentially, each compound was corrected in run-day blocks, by registering the medians to equal one, and normalising each data point proportionately. Each biochemical data in OrigScale data was then rescaled, to set the median equal to 1. Compounds in which more than 50% of values were missing were not included in the statistical analyses. Scaled data are provided in Supplementary Data 2 and 15. Raw and OrigScale data for VP are provided in Supplementary Data 16 and 17. Raw serum data are provided in Supplementary Data 18. These tables include RI, accurate mass values, mean differences in the detected metabolite, and conversion to parts per million (PPM). Metabolomic data were log-transformed (applying the natural logarithm to the data plus one) before data analysis.

*Data analysis*: Principal Component Analysis (PCA) using R software was used to visualise the metabolomic data. Before PCA, data were imputed using a *k*-nearest neighbour (*k*NN) algorithm[57] (with *k* = 5); they were then mean-centered and scaled to unit variance. Two-way ANOVA was used to compare the diets (irrespective of genotypes) or genotypes (irrespective of diets) and a *t* test was used for two groups' comparison (Supplementary Data 2 and Supplementary Data 15). Differences were considered significant when *P* was <0.05; and to account for multiple testing, a FDR[58] of <0.15. Qlucore Omics Explorer (http://www.qlucore.com; version 3.1) was used for heatmap representation and unsupervised clustering of metabolites that were significantly altered by HFD in a WT or a MYC context, or by MYC overexpression irrespective of the diet. Metabolites were grouped into 8 different classes (lipids, aminoacids, nucleotides, peptides, carbohydrates, cofactors and vitamins, energy, or xenobiotics), according to Metabolon's classification. Biochemical annotations were assigned by PhD level biochemists at Metabolon, integrating information from literature and public databases (e.g. HMDB). Metabolite Set Enrichment Analysis (MSEA) was performed using a hand-curated metabolite set (Supplementary Data 19) and run using the Gene Set Enrichment Analysis platform (GSEA; Broad Institute)[33] using 1000 permutations. Metabolite sets including fewer than three metabolites were excluded from the analysis. Metabolite sets were considered significantly enriched at *P* < 0.05 and FDR < 0.15.

**Global chromatin profiling**. The global chromatin profiling assay was performed as described in Creech et al.[59], with the following modifications:

*Cell lysis, tissue lysis, and histone extraction*: Flash-frozen tissue samples, 10–40 mg in mass, were thawed on ice and resuspended in 200 μL ice-cold PBS. Samples were homogenised for about 2 min using a motorised pestle (VWR, 47747–370), and were spun down at 4 °C, at 1500 g for 5 min. Supernatant was removed and 0.5 mL ice-cold nucleus buffer was added to the resultant pellet. Nuclei were centrifuged at 4 °C, at 10,000 g for 1 min and supernatant was removed. The nucleus isolation procedure was repeated twice, removing supernatant each time. Histones were extracted from the remaining pellet, with 400 μL 0.4 N $H_2SO_4$ at room temperature for 16 h, while shaking; at this point, histone isolation proceeded using the same protocol as described[59]. In addition to the flash-frozen tissue, histones were extracted from one 25 million cell pellet each of Arg-$^{15}N_6$,$^{13}C_4$ SILAC-labeled HeLa, K562, and 293 T (as in Jaffe et al.[29]), following the protocol described by Creech et al.[59].

*Histone derivatization*: The sample set used SILAC standardisation, with histones extracted from HeLa, K562, and 293T cell lines, as described above. In this workflow, input amount was reduced to 10 μg per sample (5 μg sample and 5 μg SILAC heavy standards), based on the protocol. Samples were adjusted to 100 mM sodium phosphate, pH 8.0, by adding 3 μL 500 mM sodium phosphate, pH 8.0; the total volume of the sample was brought up to 15 μL with HPLC-grade water. Phosphate-buffered samples were reacted with 60 μL of 400 mM NHS propionate in anhydrous methanol at room temperature, with shaking. Three hundred microliters of 0.1% trifluoroacetic acid (TFA) was added, to bring samples to a volumetric concentration of 20% organic solvent. Samples were desalted on a 96-well Oasis HLB 5 mg/cc plate (Waters, 186000309). Activation, equilibration, and wash volumes were 200 μL for each step, and sample elution volume was 100 μL. For the trypsin digestion, 1 μg trypsin was used in 10 μL of 50 mM ammonium bicarbonate, pH 8.0, while all other conditions were as described[59]. After digestion and lyophilization, new N-termini were derivatized, by resuspending peptides in 40 μL of 400 mM NHS propionate/anhydrous methanol, and adjusting to 18 mM sodium phosphate, pH 8.0, with 10 μL 100 mM sodium phosphate, pH 8.0. The reaction was quenched with 10 μL 15% hydroxylamine solution and incubated for 30 min at room temperature with shaking. Samples were brought up to a total volume of 260 μL with HPLC-grade water, frozen, and lyophilised via vacuum concentrator. Samples were resuspended in 200 μL 0.1% TFA, and desalted on a SepPak tC18 96-well μElution plate (Waters, 186002318). All activation and wash volumes were 200 μL. Elution volume was 100 μL. Desalted peptides were lyophilised via vacuum concentrator, and were brought up to a volume of 10 μL with 3% acetonitrile (ACN)/5% formic acid (FA). Samples were further diluted 1:10 with 3% ACN/5% FA, before introducing them into the mass spectrometer.

*LC-MS/MS assay parameters*: The gradient was modified so that peptides were separated at a flow rate of 200 nL/minute, with a 60 min linear gradient from 97% solvent A (3% ACN/ 0.1% FA) to 33% solvent B (90% ACN/ 0.1% FA). This gradient was followed by a 15 min linear gradient, from 33% solvent B to 65% solvent B. This gradient was followed by a 5 min linear gradient from 65% solvent B to 90% solvent B, at which point the 90% solvent B was held for an additional 5 min. Including sample loading and column equilibration times, each sample took 120 min to completion, 90 min of which was taken up by active data acquisition.

*Scheduling for H3, H4, H2A, H2A.Z and H2B targets*: To determine each peptide's retention time, we employed a scheduling sample, comprising three samples in a 1:1:1 ratio instead of a synthetic peptide mix. Most method parameters were the same as in Creech et al.[59], except that peptides were scheduled within a 23-min window, based on hypothesised elution time; also, the total run time for each scan was 0–90 min. The isolation width for MS1 and MS2 scans were narrowed to 1.7 m/z with a 0.3 m/z offset: these data were acquired on a Q-Exactive Plus (Thermo Scientific) mass spectrometer. A list of peptides targeted in addition to published histone marks in Creech et al.[59] is presented in Supplementary Data 20.

*Scheduled data acquisition*: After determining retention times, 1 μL of sample was injected onto the same column that was utilised for scheduling, using the same gradients with previously described modifications. MS1 and MS2 scans used the same parameters as described in Creech et al.[59], with the same scan run time and isolation width modifications as described above. The inclusion list was turned on for each MS2 scan, and included heavy as well as light versions of each peptide to be observed, its charge state, new acquisition windows based on the scheduling runs, and optimal collision energies.

*Heatmap generation*: GENE-E (http://www.broadinstitute.org/cancer/software/GENE-E/) was used for heatmap representation as well as statistical analysis of the data, using the comparative marker selection suite[60]. Differences were considered significant if the *p*-value was <0.05, and FDR was <0.1. Unsupervised clustering of histone marks (one minus Pearson correlation) was done on normalised values, based on the median level of each mark in the three WT prostate lobes (VP, DLP and AP).

**ChIP-sequencing**. The ChIP-sequencing was performed as described in Ku et al.[61], with the following modifications. Fresh-frozen VP tissues from 12-week-old mice were pulverised (Cryoprep Impactor, Covaris), resuspended in PBS + 1% formaldehyde, and incubated at room temperature for 20 min. Fixation was stopped by the addition of 0.125 M glycine (final concentration) for 15 min at room

temperature, then washing in ice-cold PBS + EDTA-free protease inhibitor cocktail (PIC; #04693132001, Roche). Multiple biological replicates were combined for each condition in two distinct pools (replicates). Chromatin was isolated by the addition of lysis buffer (0.1% SDS, 1% Triton X-100, 10 mM Tris-HCl (pH 7.4), 1 mM EDTA (pH 8.0), 0.1% NaDOC, 0.13 M NaCl, 1X PIC) + sonication buffer (0.25% sarkosyl, 1 mM DTT) to the samples, which were maintained on ice for 30 min. Lysates were sonicated (E210 Focused-ultrasonicator, Covaris) and the DNA was sheared to an average length of ~ 200–500 bp. Genomic DNA (input) was isolated by treating sheared chromatin samples with RNase (30 min at 37 °C), proteinase K (30 min at 55 °C), de-crosslinking buffer (1% SDS, 100 mM NaHCO3 (final concentration), 6–16 h at 65 °C), followed by purification (#28008, Qiagen). DNA was quantified on a NanoDrop spectrophotometer, using the Quant-iT High-Sensitivity dsDNA Assay Kit (#Q33120, Thermo Fisher Scientific). On ice, ChIP-validated H4K20me1 (2 μg, #ab9051, Abcam) or PHF8 (5 μg, #A301–772A, Bethyl Laboratories) antibodies[62] were conjugated to a mix of washed Dynalbeads protein A and G (Thermo Fisher Scientific), and incubated on a rotator (overnight at 4 °C) with 1.5 μg (H4K20me1) or 5 μg (PHF8) of chromatin. ChIP'ed complexes were washed, sequentially treated with RNase (30 min at 37 °C), proteinase K (30 min at 55 °C), de-crosslinking buffer (1% SDS, 100 mM NaHCO3 (final concentration), 6–16 h at 65 °C), and purified (#28008, Qiagen). The concentration and size distribution of the immunoprecipitated DNA was measured using the Bioanalyzer High Sensitivity DNA kit (#5067–4626, Agilent). Dana-Farber Cancer Institute Molecular Biology Core Facilities prepared libraries from 2 ng of DNA, using the ThruPLEX DNA-seq kit (#R400427, Rubicon Genomics), according to the manufacturer's protocol; finished libraries were quantified by the Qubit dsDNA High-Sensitivity Assay Kit (#32854, Thermo Fisher Scientific), by an Agilent TapeStation 2200 system using D1000 ScreenTape (# 5067–5582, Agilent), and by RT-qPCR using the KAPA library quantification kit (# KK4835, Kapa Biosystems), according to the manufacturers' protocols; ChIP-seq libraries were uniquely indexed in equimolar ratios, and sequenced to a target depth of 40 M reads on an Illumina NextSeq500 run, with single-end 75 bp reads; Bowtie2 (version 2.2.1) was used to align the ChIP-seq datasets to build version NCB37/MM9 of the mouse genome[63]. Alignments were performed using default parameters that preserved reads mapping uniquely to the genome without mismatches.

**H4K20me1**. H4K20me1 read density between transcriptional start site (TSS) and transcriptional end site (TES) was averaged for each gene, and reported against the CTD_WT (reference) for the HFD_WT, CTD_MYC or HFD_MYC conditions. Waterfall plots of rank-ordered $log_2$-fold changes were used to visualise H4K20me1 dynamic changes. Genes with a loss (<1.15 fold-change) or a gain (>1.15 fold-change) of the H4K20me1 mark between TSS and TES relative to the CTD_WT (reference) were identified for the HFD_WT, CTD_MYC and HFD_MYC conditions, and were associated with their corresponding transcript abundance. Venn diagrams were generated using the 'VennDiagram' R package (version 1.6.9).

**RNA-sequencing**. Fresh VP tissues from 12-week-old mice were dissociated to form a single cell suspension. RNA from a similar number of cells was extracted using the miRNeasy Micro Kit (#217084, Qiagen) coupled with on-column DNAse treatment (#79254, Qiagen). RNA sample concentration was measured and subjected to quality evaluation, using a Bioanalyzer RNA 6000 Nano kit (#5067–1511, Agilent). The Dana-Farber Cancer Institute Molecular Biology Core Facilities prepared libraries from 500 ng of purified total RNA, using TruSeq Stranded mRNA sample preparation kits (#RS-122–2101, Illumina) according to the manufacturer's protocol; submitted the finished libraries to quality control analyses as described in the ChIP-seq Methods section, pooled uniquely indexed RNA-seq libraries in equimolar ratios, and sequenced these to a target depth of 40M reads on an Illumina NextSeq500 run with single-end 75 bp reads. Fastq files were aligned to the mm9 genome using tophat with default parameters (version 2.0.11). Transcript abundances were calculated using the cuffquant module of Cufflinks (version 2.2.0). FPKM values were calculated and normalised using the cuffnorm module of Cufflinks (version 2.2.0). Paired *t*-test was calculated using the t.test function in R (version 3.3.2).

**Murine gene set enrichment analysis and MYC signature**. Gene expression values from biological triplicates were input for Gene Set Enrichment Analysis (GSEA)[33] using the Hallmark (H, v5.01; Supplementary Data 7) or the Chemical and Genetic Perturbations (C2.cgp, v5.1; Supplementary Data 8) Molecular Signature Databases (MSigDB) with 10,000 permutations. The Normalised Enrichment Score (NES)—associated with gene sets that were significantly enriched or depleted ($p < 0.05$ and FDR < 0.1)—was used for heatmap generation, using a custom-made R script. A murine prostatic MYC signature was obtained by combining leading edge genes from all MYC-related gene sets that were significantly enriched ($P < 0.05$ and FDR < 0.1) in the H and C2.cgp MSigDB (Supplementary Data 9). Aggregate read density profiles of PHF8 and H4K20me1, and their quantification around MYC signature genes, were generated using deepTools[64]. Mapped regions were visualised using the Integrated Genomics Viewer (IGV, version 2.3.68)[65].

**Protein analysis**. Fresh-frozen VP tissues from 12-week-old mice were pulverised (Cryoprep Pulvrizer, Covaris) and lysed on ice in RIPA buffer (20 mM Tris-HCl pH 7.5, 150 mM NaCl, 1 mM EDTA, 1 mM EGTA, 1% NP-40) with the addition of phosphatases and protease inhibitor cocktail tablets (Complete Mini, EDTA-free, Roche). MYC-CaP cells (kindly provided by Dr. Charles Sawyers, Memorial Sloan Kettering Cancer Center, New York, NY)[66] were rinsed on ice with PBS and lysed as for the mouse prostates. Cells were authenticated via STR profiling (DDC Medical, 16 January 2015). Cells were tested negative for mycoplasma contamination using MycoAlert™ Mycoplasma Detection Kit (Lonza). Equal amounts of protein (15–20 μg; Bradford protein assay, Bio-Rad) were resolved on precast 4–12 or 4–20% Tris-glycine SDS-polyacrylamide gels (Invitrogen), and transferred to Nitrocellulose Blotting membranes (Amersham), following standard procedures. Membranes were probed with the following antibodies according to the manufacturer's instructions: rabbit monoclonal [Y69] anti-c-MYC (#ab32072, Abcam), or rabbit polyclonal anti-β-Actin (#4967, Cell Signaling Technology). Densitometry analyses were made with ImageJ (U.S. NIH, Bethesda, MD; http://imagej.nih.gov/ij/). Results were normalised to β-actin and expressed as arbitrary units.

**Epidemiological studies**. *Study population*: We tested our hypothesis among prostate cancer patients who were enrolled in two prospective studies: the Physicians' Health Study (PHS) and the Health Professionals Follow-up Study (HPFS). PHS I and II began in 1982 and 1997, respectively, as randomised trials of aspirin (PHS I) and dietary supplements (PHS II), and enrolled 29,067 male U.S. physicians for the primary prevention of cardiovascular disease and cancer[67–70]. The HPFS was initiated in 1986, when 51,529 U.S. men, 40–75 years of age and working in health professions, completed a biennial questionnaire mailed to them[71]. In both studies, participants were followed by means of regular questionnaires, and self-reported data on diet, lifestyle behaviours, medical history, and disease outcomes were collected. We confirmed the incidence of prostate cancer cases in this population by reviewing medical records and pathology reports. Following the confirmation of diagnosis, we retrieved archival formalin-fixed paraffin-embedded (FFPE) prostate tissue specimens, collected during radical prostatectomy or transurethral resection of the prostate. Pathologists undertook a standardised histopathologic review, including Gleason grading[72], and standardised clinical data were abstracted from medical records. Deaths were ascertained via mail, telephone, and through periodic systematic searches of the National Death Index. Lethal prostate cancer was defined as the occurrence of distant metastases, or death due to prostate cancer. Men were followed through March 2011 for PHS and through December 2011 for HPFS. We obtained written informed consent from all participants, and the study was approved by institutional review boards at the Harvard T.H. Chan School of Public Health and Partners Health Care.

*Whole-transcriptome expression profiling*: In the current study, we undertook gene expression profiling of archival tumour tissue among 402 men with prostate cancer in the cohorts using an extreme case control design. Cases were men with lethal prostate cancer (developed metastatic disease or died from prostate cancer) and controls were men with indolent cancer (those survived at least 8 years after prostate cancer diagnosis, without any evidence of metastases). In total, there were 113 lethal cases and 289 indolent cases. We also included adjacent normal tissue for a subset of these tumour tissues (n = 200). Gene expression profiling of archival FFPE tissue was performed as described[73]. Briefly, two to three 0.6-mm cores were sampled from regions of high-density tumour, and from adjacent normal prostate tissue. RNA was extracted with the Agencourt FormaPure kit (Beckman Coulter), with use of the Biomek FX[P] automated platform. Whole-transcriptome amplification was performed using WT-Ovation FFPE System V2 (NuGEN) and the amplified cDNA was hybridised to a GeneChip Human Gene 1.0 ST microarray (Affymetrix). For the expression profiles generated, we regressed out technical variables and then shifted the residuals to derive the original mean expression values, and normalised these using the robust multi-array average method[74,75]. NetAffx annotations were used to map gene names to Affymetrix transcript cluster IDs, as implemented in the Bioconductor annotation package pd.hugene.1.0.st.v1; this resulted in 20,254 unique gene names.

*Diet assessment*: Self-administered semi-quantitative food frequency questionnaires (FFQs) were collected every four years from 1986 for the HPFS, and were administered once between 1999 and 2002 for the PHS. The FFQs asked men to report their usual intake of approximately 130 foods and beverages during the previous year, and also their fried food consumption, the type of cooking fat they used, and whether they consumed the visible fat on meat. Fat intake levels were estimated by multiplying the frequency of intake by the amount of the fat in the specific portion of each food (based on nutrient composition data from the US Department of Agriculture, supplemented with food manufacturer data), and were summed across all foods. The FFQ was validated among 127 men in the HPFS. The correlations between the FFQ and four prospectively collected one-week weighed diet records were 0.67 for total fat, and 0.75 for saturated fat[76]. Because FFQ was mainly administered after the diagnosis of prostate cancer for the PHS participants, we estimated post-diagnostic fat intakes in both HPFS and PHS, to maintain a bigger sampler size and harmonize the two cohorts. In HPFS, we calculated cumulative average post-diagnostic intake from the FFQ preceding diagnosis until the end of the follow-up in HPFS[77]. Fat intake (g/d) was multiplied by 9 kcal and divided by total calories per day to calculate the percent of daily calories from each fat of interest.

*Statistical analysis*: Fat intake after diagnosis was estimated in 4577 men enrolled in the HPFS and in 926 men from the PHS, all of whom had non-metastatic prostate cancer. Cohort-specific quintiles were determined based on fat intake distributions for each cohort, with the highest quintile denoted as the high-fat group and the lower four quintiles grouped as the low-fat group (Supplementary Data 21). The categorised fat intake groups were then integrated with gene expression data in tumour or in adjacent normal tissues. Finally, we had 319 tumour tissues from patients (213 from the HPFS and 106 from the PHS) for whom we had complete fat intake estimation (animal fat: high-fat group n = 65 vs. low-fat group n = 254; saturated fat: high-fat group n = 62 vs. low-fat group n = 257; monounsaturated fat: high-fat group n = 66 vs. low-fat group n = 253; polyunsaturated fat: high-fat group n = 55 vs. low-fat group n = 264) and a total of 157 adjacent normal tissues after merging with fat intake data (animal fat: high-fat group n = 33 vs. low-fat group n = 124; saturated fat: high-fat group n = 29 vs. low-fat group n = 128; monounsaturated fat: high-fat group n = 33 vs. low-fat group n = 124; polyunsaturated fat: high-fat group n = 24 vs. low-fat group n = 133).

*Gene set enrichment analysis*: Gene expression profiles of tumour and adjacent normal prostate tissues were input for GSEA[33], with use of the Hallmark (H, v4.0) MSigDB with 10,000 phenotype-based permutations, to identify predefined sets of functionally related genes correlated with specific fat intakes (Supplementary Data 10, 12–14). Gene sets with P < 0.05 and FDR < 0.1 were considered for subsequent analyses. Animal fat and saturated fat intake-dependent MYC signatures were obtained by combining either the leading edge or the non-leading edge genes from the MYC_targets_V1 gene set from the H MSigDB in tumour tissues (Supplementary Data 11), to create a metagene score as previously described[78]. This was computed for each sample by averaging the normalised (mean-centered and variance scaled) expression values of all member genes. An additional signature was derived from 113 randomly selected genes from the MYC_targets_V1 gene set (Supplementary Data 11). Odds ratios and 95% confidence intervals were obtained by logistic regression for the association between the metagene score and lethal prostate cancer. The score was modelled as categorical (tertiles). We tested for linear trend across score categories by modelling the tertiles as a continuous variable. All models were adjusted for age and year at diagnosis. We further adjusted for Gleason grade to test whether the score is an independent predictor of lethal prostate cancer and BMI at diagnosis, to differentiate the effect from overweight/obesity. To assess whether the association between the score and lethal prostate cancer was modified by saturated fat intake, we obtained P for interaction by including an interaction term (saturated fat intake x MYC score) in the multivariable model using a Wald test. All analyses were conducted using SAS version 9.3 and R version 3.1.0.

*Validation cohorts*: To investigate the power of SFI-induced and non-SFI-induced MYC signatures to predict metastatic disease, we utilised genome-wide expression profiles of 751 patients with metastatic outcome follow-up from the Decipher Genomic Resource Information Database (GRID; NCT02609269). These patients were pooled from four studies of either case-cohort or cohort design. Patients for these studies came from four institutes: Thomas Jefferson University (TJU; n = 139)[79], Johns Hopkins Medical Institutions-I (JHMI-I; n = 260)[80], Mayo Clinic (n = 235)[81], Cedars-Sinai (n = 117)[82]. A total of 120 non-randomly selected patients from case-cohort studies were removed before pooling the studies to avoid bias in estimating the hazard ratio. 631 patients were thus eligible for analysis, 70 of which developed metastasis. Median follow-up time for censored patients was 8 years and the median age at radical prostatectomy was 61 years.

The fat-induced MYC signature (113 genes) and non-fat-induced MYC signature (87 genes) were used to calculate pathway expression scores for each patient, using a z-score scaled, mean gene expression. Based on the tertiles of these scores, patients were divided into three groups with T1 being the lowest and T3 the highest. Kaplan–Meier curves and Cox proportional hazard regression were used to evaluate the metastatic prognosis. To test associations between signatures and BMI, we extracted BMI data from 494 patients pooled from three cohorts (TJU, n = 139; JHMI-I only, n = 144; JHMI-II[83] only, n = 95; JHMI-I/II, n = 116). Correlation analysis using Pearson's correlation was used to measure the association between MYC signatures score and BMI. JHMI-II was excluded from the survival analysis because only patients that developed biochemical recurrence were selected for this study, hence it was statistically inappropriate to pool the JHMI-II cohort with the others lacking this inclusion criteria as it would inflate the event rate. We also conducted univariate and multivariate analyses to associate the SFI-induced MYC signature with clinical outcome after adjusting for other clinicopathologic variables including pre-operative prostate-specific antigen (PSA) levels, seminal vesicle invasion, surgical margins, extracapsular extension, lymph node invasion, gleason grade or the Cell Cycle Progression score in the pooled cohort from which we utilised genome-wide expression profiles of 631 patients (deidentified and aggregated from routine clinical use of the Decipher prostate cancer classifier test; Decipher Biosciences Laboratory, San Diego, CA) with metastatic outcome follow-up from the Decipher GRID.

**Adequacy of statistical analyses**. All the statistical tests were justified as appropriate. Assumption criteria were met, analysis of variance was performed. When variance was not equal, Welch's t-test (unequal variance t-test) was applied. Data are reported including estimation of variation within each group. Two-sided tests were used. Measurements were taken from distinct samples.

**Reporting summary**. Further information on research design is available in the Nature Research Reporting Summary linked to this article.

## Data availability

Data are available from the corresponding authors upon request. Raw data underlying reported averages in graphs and uncropped versions of blots are provided in the source data file or supplementary tables. Raw metabolomics data generated by Metabolon were deposited on MetaboLights and are available through the study identifier MTBLS135. Raw, scaled metabolomics data, and statistics were also provided as supplementary tables. The sequencing data reported in this paper (ChIP-seq and RNA-seq) were deposited on NCBI Gene Expression Omnibus (GEO) and are accessible through GEO Series accession number GSE90912. Human gene expression data is available through GSE62872 [https://www.ncbi.nlm.nih.gov/geo/query/acc.cgi].

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

## Acknowledgements

We thank Zach Herbert, Julie Batista, Noriko Uetani, Cornelia Photopoulos, Radha Kalekar, Maura Cotter, Natalia Scaglia, Teresa Bowman, Nadia Boufaeid (technical assistance and/or helpful discussions), and Sonal Jhaveri (critical review of the manuscript). D.P.L. is a Lewis Katz—Young Investigator of the Prostate Cancer Foundation, and is the recipient of a Scholarship for the Next Generation of Scientists from the Cancer Research Society, and is also recipient of a Canadian Institute of Health Research Fellowship. G.Z. is a recipient of the Idea Development Award from the U.S. Department of Defense (DoD; PC150263) and the Barr Award from the Dana-Farber Cancer Institute. C.Y.L. is a CPRIT Scholar in Cancer Research (RR150093), a Pew-Stewart Scholar for Cancer Research (Alexander and Margaret Stewart Trust), and is funded by the NIH and NCI (1R01CA215452-01). E.M.E. is supported by the DoD Prostate Cancer Research Program Postdoctoral Training Award (W81XWH-14-1-0250). L.E., D.E.S. and L.A.M. are Young Investigators of the Prostate Cancer Foundation. Support for HPFS/PHS cohorts was provided by grants from the DoD (W81XWH-11-1-0529) and grants from the National Institute of Health (NIH; CA42182, CA58684, CA90598, CA141298, CA97193, CA34944, CA40360, CA131945, CA167552, P50CA090381, 1U54CA155626-01, P30DK046200, HL26490 and HL34595). We would like to thank the participants and staff of the HPFS/PHS for their valuable contributions as well as the following state cancer registries for their help: A.L.Z., A.Z., A.R., C.A., C.O., C.T., D.E., F.L., G.A., I.D., I.L., I.N., I.A., K.Y., L.A., M.E., M.D., M.A., M.I., N.E., N.H., N.J., N.Y., N.C., N.D., O.H., O.K., O.R., P.A., R.I., S.C., T.N., T.X., V.A., W.A., W.Y. We assume full responsibility for analyses and interpretation of these data. The work reported here was supported by grants from the NIH (R01CA131945, R01CA187918 to M.L. and P50CA090381 to P.W. K., M.L. and M.B.), the NCI (1P01CA163227 to M.B.) and the Prostate Cancer Foundation to M.L. and M.B.

## Author contributions

D.P.L., G.Z., M.L. and M.B. conceived the study and designed the experiments. Wet lab experiments were performed by D.P.L., G.Z. and A.L.C. and supported by H.E., S.S., L.E., J.D.J., M.L. and M.B. E.D.K. performed MS-based metabolomics analysis. Computational analyses were performed by D.P.L., M.Y., J.M.R., C.Y.L., S.C., M.A. and N.E. and supported by E.M.E., E.A.G., M.T., J.L., E.D., A.V.D., P.W.K., J.E.B., L.A.M., J.E.C., M.L. and M.B. Access to validation cohorts data was provided by A.R., E.M.S., R.B.D., R.J.K., S.J.F. and D.E.S. Pathological analyses were performed by F.G., M.F. & M.L. D.P.L., G.Z., M.Y., S.C., A.L.C., M.L. and M.B. interpreted the data and drafted the paper considering inputs from all co-authors.

## Additional information

**Competing interests:** C.Y.L. receives sponsored research from and consults for Kronos Bio, is a shareholder and inventor of IP licensed to Syros Pharmaceuticals, is a shareholder of Amgen, and is an equity partner of Cambridge Science Corporation. M.A., N.E., M.T., J.L., E.A.G. and E.D. are employees of Decipher Biosciences. E.D.K. is currently employed of Metabolon. M.B. receives sponsored research support from Novartis. M.B. is a consultant to Aleta Biotherapeutics and H3 Biomedicine and serves on the SAB of Kronos Bio. The remaining authors declare no competing interests.

David P. Labbé [1,2,23,25], Giorgia Zadra [3,4,25], Meng Yang[5], Jaime M. Reyes [1], Charles Y. Lin[6], Stefano Cacciatore [7], Ericka M. Ebot[8], Amanda L. Creech[9], Francesca Giunchi[10], Michelangelo Fiorentino[10], Habiba Elfandy[3], Sudeepa Syamala[3], Edward D. Karoly[11], Mohammed Alshalalfa[12], Nicholas Erho[12], Ashley Ross[13], Edward M. Schaeffer[14], Ewan A. Gibb[12], Mandeep Takhar[12], Robert B. Den[15], Jonathan Lehrer[12], R. Jeffrey Karnes[16], Stephen J. Freedland[17,18], Elai Davicioni[12], Daniel E. Spratt[19], Leigh Ellis [3,4,9], Jacob D. Jaffe [9], Anthony V. D'Amico[20], Philip W. Kantoff[1,21], James E. Bradner [1], Lorelei A. Mucci[8,22], Jorge E. Chavarro[5,8,22], Massimo Loda[3,4,9,24] & Myles Brown[1,2]

[1]Department of Medical Oncology, Dana-Farber Cancer Institute, Harvard Medical School, Boston, MA, USA. [2]Center for Functional Cancer Epigenetics, Dana-Farber Cancer Institute, Boston, MA, USA. [3]Department of Oncologic Pathology, Dana-Farber Cancer Institute, Boston, MA, USA. [4]Department of Pathology, Brigham and Women's Hospital, Boston, MA, USA. [5]Department of Nutrition, Harvard T.H. Chan School of Public Health, Boston, MA, USA. [6]Department of Molecular and Human Genetics, Baylor College of Medicine, Houston, TX, USA. [7]Cancer Genomics Group, International Centre for Genetic Engineering and Biotechnology, Cape Town, South Africa. [8]Department of Epidemiology, Harvard T.H. Chan School of Public Health, Boston, MA, USA. [9]The Broad Institute of MIT and Harvard University, Cambridge, MA, USA. [10]Pathology Service, Addarii Institute of Oncology, S-Orsola-Malpighi Hospital, Bologna, IT, Italy. [11]Metabolon, Morrisville, NC, USA. [12]Decipher Biosciences, Vancouver, BC, Canada. [13]James Buchanan Brady Urological Institute, Johns Hopkins Medical Institutions, Baltimore, MD, USA. [14]Northwestern University Feinberg School of Medicine, Chicago, IL, USA. [15]Department of Radiation Oncology, Sidney Kimmel Medical College at Thomas Jefferson University, Philadelphia, PA, USA. [16]Department of Urology, Mayo Clinic Rochester, Rochester, MN, USA. [17]Department of Surgery, Division of Urology, Center for Integrated Research on Cancer and Lifestyle, Samuel Oschin Comprehensive Cancer Center, Cedars-Sinai Medical Center, Los Angeles, CA, USA. [18]Surgery Section, Durham Veteran Affairs Medical Center, Durham, NC, USA. [19]Department of Radiation Oncology, University of Michigan, Ann Arbor, MI, USA. [20]Department of Radiation Oncology, Dana-Farber Cancer Institute and Brigham and Women's Hospital, Harvard Medical School, Boston, MA, USA. [21]Department of Medicine, Memorial Sloan Kettering Cancer Center, New York, NY, USA. [22]Channing Division of Network Medicine, Brigham and Women's Hospital, Harvard Medical School, Boston, MA, USA. [23]Present address: Division of Urology, Department of Surgery, McGill University and Research Institute of the McGill University Health Centre, Montréal, QC, Canada. [24]Present address: Department of Pathology and Laboratory Medicine, Weil Cornell Medicine, New York Presbyterian-Weill Cornell Campus, New York, NY, USA. [25]These authors contributed equally: David P. Labbé, Giorgia Zadra.

