## [Peer Review File · Nature Communications]

Reviewers' comments:

Reviewer #2 (Remarks to the Author):

This Labbe et al. revised study is aimed at characterizing the molecular and metabolic alterations that result from increased consumption of saturated fat (or high-fat diet (HFD)), MYC signaling and that are associated with increased risk of prostate cancer progression. The original submission was over a year ago and the authors have added new analyses of the data from the original submission. The authors have responded to all critique points with either a discussion/clarification and/or new analyses or presentation of the data (e.g. Fig 2c,e; Fig. 4a,b,e) that was part of the original submission. While clarifications and new figure panels were responsive to some of the comments, there are issues related to this reviewers original critique that still need to be addressed.

1. The counterintuitive result regarding lower h3k27me3 but higher Ezh2 RNA needs to be shown in the figure and discussed in text. Many readers may want to know this result. It could be that H3K27me3 mark and EZH2 activity is redirected in the face of HFD.

2. As noted in the first critique, the difference between H4K20me status in MYC CTD vs HFD curves (Fig 2d) is modest. Also, even though a significant p value is being reported the degree to the gain in gene expression in MYC HFD vs MYC CTD (Fig2e) is modest at best. The same comment applies for the HFD vs CTD enrichment of V1/V2 MYC signature (Fig 3b). The author agrees and in the rebuttal noted that these signatures are enriched in CTD MYC vs CTD WT to the same degree as HDF MYC vs HFD WT. The text needs to be changed to indicate that these signatures are not further enriched in MYC HFD tumors. Also, is this modest change expected? If so, why?

3. HFD was associated with increased PHF8 occupancy and associated H4K20 demethylation at MYC targets. Was this specific to MYC targets or was this induction a general effect in MYC over-expressing, HFD mouse models? Where other gene sets impacted? A summary of genome-wide occupancy in the 4 conditions needs to be reported in order for the reader to interpret results in Fig 3c-e. The authors comment that **[REDACTED]**

4. In the author's response to the comment regarding the cited Hi-MYC model^{6,9} specific signaling molecules and pathways, it was noted that of all of the pathways/molecules evoked in the other reports, ... **[REDACTED]**

Even this study was using younger mice, before onset of HFD-induced phenotype, this needs to be noted in the manuscript to allow the reader to put this result into perspective.

Reviewer #3 (Remarks to the Author):

My concerns have largely been addressed. I still can not endorse publication of this manuscript as

a key aspect to it is the metabolomics data which was acquired with a for-profit company with insufficient detail about how the experiments were performed. If the authors can repeat the metabolomics data in an academic setting and provide clear reference to how the data were collected, I would be happy to give this manuscript further consideration.

Reviewer #5, Replacement Reviewer for Reviewer #1 and #4 (Remarks to the Author):

The authors have undertaken additional analyses to resolve points raised about the disconnects between the mouse and human studies.

The revised manuscript should more fully reflect this work. For example it should include a direct comparison of the Myc signature versus Myc expression/staining in the Harvard cohort. It should include a direct comparison of the Myc signature with the Prolaris signature but also with the Decipher signature. Presumably the Myc signature is more robust than Myc itself in prognosticating these cohorts because the genes within the Myc signature are also reflecting other poor prognosis oncogenic drivers? Given the heterogeneity of prostate cancers that would seem likely. The authors should entertain this possibility in the discussion and if possible comment on it in the light of the differential expression of other known oncogenes and tumour suppressors within the clinical datasets. The essence of the disconnect in the manuscript is that the original signature reflects the influence of diet on a 'pure' Myc transgenic model. Once you move into the human disease you de facto sacrifice the luxury of defining precisely the oncogenic driver. Consequently the influence of a high-fat diet is more likely here to reflect the convergence of oncogenic signalling pathways on dysregulated lipid metabolism in prostate cancer. The authors need to reflect this more fully in the paper.

On a more minor note, signatures are best defined by a weighted score for each contributory gene and tend to most robust when reflecting admixtures of differentially up- and down-regulated genes. Given the number of genes in the murine signature (610) and the number of leading edge genes in the human signature (122), it would be helpful to have some indication of the relative contribution of these to the prognostication of the cases.

Second Rebuttal

Reviewer #2:

Remarks to the Author:

This Labbé et al. revised study is aimed at characterizing the molecular and metabolic alterations that result from increased consumption of saturated fat (or high-fat diet (HFD)), MYC signaling and that are associated with increased risk of prostate cancer progression. The original submission was over a year ago and the authors have added new analyses of the data from the original submission. The authors have responded to all critique points with either a discussion/clarification and/or new analyses or presentation of the data (e.g. Fig 2c,e; Fig. 4a,b,e) that was part of the original submission. While clarifications and new figure panels were responsive to some of the comments, there are issues related to this reviewers original critique that still need to be addressed.

1. The counterintuitive result regarding lower h3k27me3 but higher Ezh2 RNA needs to be shown in the figure and discussed in text. Many readers may want to know this result. It could be that H3K27me3 mark and EZH2 activity is redirected in the face of HFD.

Answer: We agree that this is an interesting result and it will be made publicly available to the scientific community through the **Supplementary Figure to Reviewer 1** associated to our first rebuttal.

2. As noted in the first critique, the difference between H4K20me status in MYC CTD vs HFD curves (Fig 2d) is modest. Also, even though a significant p value is being reported the degree to the gain in gene expression in MYC HFD vs MYC CTD (Fig2e) is modest at best. The same comment applies for the HFD vs CTD enrichment of V1/V2 MYC signature (Fig 3b). The author agrees and in the rebuttal noted that these signatures are enriched in CTD MYC vs CTD WT to the same degree as HDF MYC vs HFD WT. The text needs to be changed to indicate that these signatures are not further enriched in MYC HFD tumors. Also, is this modest change expected? If so, why?

Answer: We would like to reiterate the answer provided in our previous rebuttal in response to **Reviewer #2** critique. The reported data is the result of unbiased GSEA analyses, corrected for multiple comparisons (FDR<0.1). These analyses revealed that MYC overexpression (CTD_MYC) in the murine prostate lead to a clear enrichment for both MYC_targets_V1 (NES=2.847) and MYC_targets_V2 (NES=2.575) when compared to normal prostate tissues (CTD_WT). HFD in a MYC context (HFD_MYC) enriched even more for a MYC transcriptional program as shown by a MYC_targets_V1 (NES=2.879) and MYC_targets_V2 (NES=2.311) enrichment when compared to MYC overexpression alone (CTD_MYC; **Fig. 3a** and **Supplementary Table 12**) – thus supporting further enrichment of gene sets related to MYC transcriptional activity by HFD. Again, this enhanced MYC transcriptional program, as well as other transcriptional, metabolic and epigenetic alterations shown throughout the manuscript occurred in 12-week-old mice, 24 weeks before the appearance of the diet-dependent phenotype (*i.e.* increased tumor burden and proliferative index) observed in the MYC-transformed prostates (36-week-old animals). Therefore, these changes are expected to be modest.

3. HFD was associated with increased PHF8 occupancy and associated H4K20 demethylation at MYC targets. Was this specific to MYC targets or was this induction a general effect in MYC

over-expressing, HFD mouse models? Where other gene sets impacted? [...] The authors comment that

[REDACTED]

Answer: We focused our analyses on MYC targets since our experiments revealed that gene sets related to MYC transcriptional activity were the only ones further enriched by HFD in MYC-transformed murine prostates and by animal/saturated fat intake in human prostate cancer tissues. We agree that PHF8 dependence in our system via altered levels of H4K20me1 at MYC target genes is still in question and the discussion in the manuscript is worded accordingly.

4. In the author's response to the comment regarding the cited Hi-MYC model^{1, 2} specific signaling molecules and pathways, it was noted that of all of the pathways/molecules evoked in the other reports, ...

[REDACTED]

Even this study was using younger mice, before onset of HFD-induced phenotype, this needs to be noted in the manuscript to allow the reader to put this result into perspective.

Answer: Throughout the manuscript, we emphasize that the goal of the study was to evaluate HFD-induced metabolic rewiring in the prostate **before** the emergence of an HFD-dependent phenotype (increased proliferation and tumor burden; **Fig. 1c-d**). As suggested by the reviewer, these results will be available to the scientific community in the **Supplementary Figures to Reviewer 2 and 3** associated to our first rebuttal.

Reviewer #3:

Remarks to the Author:

My concerns have largely been addressed. I still can not endorse publication of this manuscript as a key aspect to it is the metabolomics data which was acquired with a for-profit company with insufficient detail about how the experiments were performed. If the authors can repeat the metabolomics data in an academic setting and provide clear reference to how the data were collected, I would be happy to give this manuscript further consideration.

Answer: To address the reviewer's concerns, the raw metabolomics data generated by Metabolon were deposited in the MetaboLights public repository. Through the MetaboLights portal maintained by the European Bioinformatics Institute (EMBL-EBI), users will be able to retrieve the raw data (including spectra) in the form of MS scan.mzml files under the study identifier MTBLS135. Details concerning access to our raw metabolomics data were also added to the "Data availability" section of our manuscript.

Reviewer #5, Replacement Reviewer for Reviewer #1 and #4 (Remarks to the Author): *The authors have undertaken additional analyses to resolve points raised about the disconnects between the mouse and human studies. The revised manuscript should more fully reflect this work. For example it should include a direct comparison of the Myc signature versus Myc expression/staining in the Harvard cohort. [...] Presumably the Myc signature is more robust than Myc itself in prognosticating these cohorts because the genes within the Myc signature are also reflecting other poor prognosis oncogenic drivers? Given the heterogeneity of prostate cancers that would seem likely. The authors should entertain this possibility in the discussion and*

if possible comment on it in the light of the differential expression of other known oncogenes and tumour suppressors within the clinical datasets.

Answer: Unfortunately, as thoroughly detailed in our first rebuttal, we faced important challenges when we attempted to retrieve and incorporate the presence of specific genetic alterations (*i.e.* oncogene ERG and tumor suppressor PTEN) to our analyses. Therefore, we acknowledge this limitation in the discussion as follows: “This suggests that the enhancement of MYC-driven metabolic and epigenetic reprogramming may be a general mechanism that underlies the influence of dietary fat intake on prostate cancer progression although this hypothesis remains to be tested across prostate cancer molecular subtypes.”

The essence of the disconnect in the manuscript is that the original signature reflects the influence of diet on a 'pure' Myc transgenic model. Once you move into the human disease you de factor sacrifice the luxury of defining precisely the oncogenic driver. Consequently the influence of a high-fat diet is more likely here to reflect the convergence of oncogenic signalling pathways on dysregulated lipid metabolism in prostate cancer. The authors need to reflect this more fully in the paper.

Answer: We would like to thank the reviewer for this very important comment and we have incorporated this observation in our discussion, as follows: “This highlights the fact that saturated fat intake not only enriches the expression of MYC-regulated genes but does so especially for the most predictive subset of genes, possibly reflecting the convergence of oncogenic signalling pathways on dysregulated lipid metabolism, a key feature for prostate cancer development and progression to a metastatic disease.³”

On a more minor note, signatures are best defined by a weighted score for each contributory gene and tend to most robust when reflecting admixtures of differentially up- and down-regulated genes. Given the number of genes in the murine signature (610) and the number of leading edge genes in the human signature (122), it would be helpful to have some indication of the relative contribution of these to the prognostication of the cases.

Answer: We believe that further analyses comparing the prognostic value of the AFI-induced murine MYC_targets_V1 signature to the AFI/SFI-induced human MYC_targets_V1 signature would add very little value to the present manuscript. Nevertheless, we have included the new evidence that AFI/SFI-induced human MYC_targets_V1 signatures (leading edge genes), but not their corresponding AFI/SFI-non-induced counterparts (non-leading edge genes) or a randomly picked MYC_targets_V1 signature, are associated with prostate cancer lethality (**Table 2** and **Supplementary Table 16**). Additional work, beyond the scope of this manuscript, will be required to identify the subset of key MYC target genes contributing to the prognosis of prostate cancer cases.

References

1. Blando J, *et al.* Dietary energy balance modulates prostate cancer progression in Hi-Myc mice. *Cancer Prev Res (Phila)* **4**, 2002-2014 (2011).
2. Kobayashi N, *et al.* Effect of low-fat diet on development of prostate cancer and Akt phosphorylation in the Hi-Myc transgenic mouse model. *Cancer Res* **68**, 3066-3073 (2008).
3. Zadra G, Loda M. Metabolic Vulnerabilities of Prostate Cancer: Diagnostic and Therapeutic Opportunities. *Cold Spring Harb Perspect Med* **8**, (2018).

Reviewers' comments:

Reviewer #2 (Remarks to the Author):

The authors provided rationale/explanation for the issues that were raised by all of the reviewers. The only minor issue for this reviewer is the authors comment to point #2 that this reviewer raised.

In this new rebuttal letter the authors state that "...These analyses revealed that MYC overexpression (CTD_MYC) in the murine prostate lead to a clear enrichment for both MYC_targets_V1 (NES=2.847) and MYC_targets_V2 (NES=2.575) when compared to normal prostate tissues (CTD_WT). HFD in a MYC context (HFD_MYC) enriched even more for a MYC transcriptional program as shown by a MYC_targets_V1 (NES=2.879) and MYC_targets_V2 (NES=2.311) enrichment when compared to MYC overexpression alone (CTD_MYC; Fig. 3a and Supplementary Table 12) – thus supporting further enrichment of gene sets related to MYC transcriptional activity by HFD."

Based on these data, there doesn't seem to be a further increase of MYC targets in the context of HFD (comparing NES = 2.847 (CTD) to 2.879 (HFD) for V1 and NES = 2.575 (CTD) to 2.311 (HFD) for V2. There is enrichment of MYC targets in MYC-expressing prostates compared to control mice but, based on these numbers, the enrichment of MYC targets in CTD vs HFD is modest at best and should be stated as such. It is not clear why the authors maintain that HFD further enriches for these targets.

Reviewer #3 (Remarks to the Author):

The essential comment I made in the previous remains unaddressed. The metabolomics data is not performed at a standard sufficient for publication.

Reviewer #5 (Remarks to the Author):

The authors have addressed my comments adequately given the constraints of time and data access as outlined in their rebuttal.

Reviewer #6 (Remarks to the Author):

I was asked to evaluate the quality of metabolomics data. Here are my comments.

1. I carefully checked the experimental details and raw data for the project (MTBLS135) in MetaboLights. However, the provided experimental details and raw data files were NOT consistent with the method descriptions in the manuscript. Therefore, I think the authors should consult an external expert on metabolomics, and re-prepare the metabolomics data.

Here are some issues (NOT ALL) about the raw data set:

1) In the uploaded data, I only found data files from LC-MS/MS platform, and no GC-MS data files were available. Please check whether all raw data files were uploaded. In Method, the authors claimed they used GC-MS for analysis (Page 23, Line 816).

2) For all uploaded LC-MS/MS data files, it included 3 different LC-MS methods: Waters BEH C18 column - acidic condition (method 1); Waters BEH C18 column - basic condition (method 3); Waters BEH Amide column (method 4). However, the method descriptions were NOT matched with the description in main text, "P23 Line 814 in main text: Extracts were divided into five fractions: one for Ultra-performance liquid chromatography tandem mass-spectrometry (UPLC-MS/MS) with positive ion mode electrospray ionization (IMEI); one for (UPLC-MS/MS) with negative IMEI; one

for liquid chromatography (LC) polar platform; and one Gas Chromatography (GC)-MS. "I assume the authors discarded the GC-MS data?"

3) The details of LC-MS methods were not provided in both MetaboLights and manuscript, for example, the elution gradients for each LC-MS method.

4) The details for metabolite identifications were missing. As a standard reporting protocol, the authors should provide a table with all identified metabolites, and indicated the following parameters:

λ Experimental m/z, library m/z., relative error

λ Experimental rt, library rt., relative error

λ MS/MS spectral similarity score

λ Positive or negative mode

2. In addition, I share the same conclusion with Reviewer 2 that the manuscript lacks many important details regarding to the metabolomics experiments, such as sample preparation, metabolite identification etc. It is worthy to note that these experimental details are not the same as the raw data. Even the authors deposited the raw metabolomics data and a part of details to the public database, it is still necessary to provide the detailed experimental descriptions both in the main text and database. Specifically, here are some examples for missing experimental details including but not limited to:

1) P23 Line 808: What were the weights of tissues? What is the recovery standard?

2) P23 Line 814: How much of methanol was added during the preparation?

3) P23 Line 820: What were the detailed parameters for UPLC-MS/MS and GC-MS analysis? Please provided the details for "with some modifications" compared to ref 26.

4) The authors should add the description of data acquisition in the main text, such as instruments, chromatography columns, solvents, mobile phases etc.

5) P23 Line 822: What was the meaning of "small volume" in the preparation of pooled sample? Please add the specific volumes.

6) P23 Line 824: What were included in cocktail of QC standards? It determined how the data quality was checked.

7) P23 Line 830: The authors should claim the used software in peak detection and integration (even it is the commercial software from Metabolon).

8) P24 Line 838: What were the spiked standards to calculate retention index?

3. P24 Line 853: In metabolite identification, the authors said "retention index within 150 RI units of the proposed identification, experimentally detected precursor mass match to the authentic standard within 0.4 m/z and a MS/MS fragmentation spectral score." Here, a tolerance of 0.4 m/z was too large for the precursor match. For Q-Exactive mass spectrometer, the typical set tolerance should be 5-10 ppm. Again, the authors used both LC-MS and GC-MS which should use two different strategies for identification. Please provide more details for metabolite identification, including the cut-off score.

4. P24 Line 878: The corrected P value (e.g. FDR) cutoff was set as less than 0.15. Why the authors set such a very loose condition?

5. P25 Line 883: How did the authors classified the metabolites? It should add some description or a reference.

6. P4 Line 136: the authors mentioned that "untargeted metabolomics identified 414 metabolites in the prostate". However, I noticed a note in Supplementary Table S3, "Compound not officially confirmed based on a standard, but confident on its identity." Therefore, the confidence level of annotations for each metabolite should be clearly given according to the definition of Metabolomics Standards Initiative (MSI). Specifically, the annotation details but not only the result of each metabolite are required to be provided, such as m/z error, RI error, MS/MS database source and MS/MS match score. It would be helpful to evaluate and confirm the accuracy and reliability of metabolite identifications.

7. A final important comment on study design, it seems that the high-fat diet has a major impact on the lipidome instead of metabolome. For example, in figure 1g, the lipid was a class of metabolites with the most significant change. Similarly, in the supplementary table 3, 53 out of 143 significantly changed metabolites were lipids. Therefore, I would suggest the authors consider an additional lipidomics experiment. It may help to discover more interesting results.

Third Rebuttal

Reviewer #2:

Remarks to the Author:

The authors provided rationale/explanation for the issues that were raised by all of the reviewers. The only minor issue for this reviewer is the authors comment to point #2 that this reviewer raised.

In this new rebuttal letter the authors state that “...These analyses revealed that MYC overexpression (CTD_MYC) in the murine prostate lead to a clear enrichment for both MYC_targets_V1 (NES=2.847) and MYC_targets_V2 (NES=2.575) when compared to normal prostate tissues (CTD_WT). HFD in a MYC context (HFD_MYC) enriched even more for a MYC transcriptional program as shown by a MYC_targets_V1 (NES=2.879) and MYC_targets_V2 (NES=2.311) enrichment when compared to MYC overexpression alone (CTD_MYC; Fig. 3a and Supplementary Table 12) – thus supporting further enrichment of gene sets related to MYC transcriptional activity by HFD.”

Based on these data, there doesn't seem to be a further increase of MYC targets in the context of HFD (comparing NES = 2.847 (CTD) to 2.879 (HFD) for V1 and NES = 2.575 (CTD) to 2.311 (HFD) for V2. There is enrichment of MYC targets in MYC-expressing prostates compared to control mice but, based on these numbers, the enrichment of MYC targets in CTD vs HFD is modest at best and should stated as such. It is not clear why the authors maintain that HFD further enriches for these targets.

Answer: We thank the reviewer for the positive comments. The authors would like to clarify the last minor issue has arisen by a misinterpretation of the NES values.

a) NES = 2.847 for V1 or 2.575 for V2 (CTD) refers to CTD_MYC vs. **CTD_WT** comparison.

b) NES = 2.879 for V1 or 2.311 for V2 (HFD) refers to HFD_MYC vs. **CTD_MYC** comparison.

It is important to note that the reference condition differs between both analyses. In a), we are demonstrating that MYC-driven prostate transformation enriches for a MYC transcriptional program (reference: **CTD_WT**). In b), we are demonstrating that a MYC transcriptional program can be further enriched when mice are fed a HFD (reference: **CTD_MYC**).

To avoid any confusion, this explanation has now been included in the legend of **Figure 3:**

High-fat diet enhances MYC transcriptional activity. (a) Gene Set Enrichment Analysis (GSEA, Hallmark, $P < 0.05$ and $FDR < 0.1$) revealed enhanced expression of MYC target genes triggered by HFD in the MYC but not in the WT context (*right column-left side*: HFD_MYC vs. CTD_MYC; *right column-right side*: HFD_WT vs. CTD_WT comparisons; *left column*: CTD_MYC vs. CTD_WT comparison).

Reviewer #3:

Remarks to the Author:

The essential comment I made in the previous remains unaddressed. The metabolomics data is not performed at a standard sufficient for publication.

Answer: To address the comments of the reviewer, we have now included a significant amount of information and details regarding the metabolomics experiments, such as sample preparation, metabolite identification and quantification, LC/MS methods, quality control, *etc.* Tables containing raw data with retention index (RI) and accurate mass information for each metabolite have been included. Detailed experimental descriptions have now been included both in the main text and online (*i.e. MetaboLights*). We think that the current presentation of the metabolomics data reaches a standard sufficient for publication and it is consistent with numerous published studies, including many published in *Nature Communications*.

Reviewer #5:

Remarks to the Author:

The authors have addressed my comments adequately given the constraints of time and data access as outlined in their rebuttal.

Answer: We thank the reviewer for the positive comment.

Reviewer #6:

Remarks to the Author:

I was asked to evaluate the quality of metabolomics data. Here are my comments.

1. *I carefully checked the experimental details and raw data for the project (MTBLS135) in MetaboLights. However, the provided experimental details and raw data files were NOT consistent with the method descriptions in the manuscript. Therefore, I think the authors should consult an external expert on metabolomics, and re-prepare the metabolomics data.*

Answer: We thank the reviewer for pointing out this problem and we have now provided a revised version of the methods section and new supplemental tables to provide the required experimental details and data consistently in both the manuscript and *MetaboLights*.

Here are some issues (NOT ALL) about the raw data set:

1) *In the uploaded data, I only found data files from LC-MS/MS platform, and no GC-MS data files were available. Please check whether all raw data files were uploaded. In Method, the authors claimed they used GC-MS for analysis (Page 23, Line 816).*

Answer: We thank the reviewer for these comments. The methods delineated in the original submission accurately described the methodology employed, however, the GC-MS data did not contribute any metabolites to the current study. Specifically, GC/MS analysis was run on these samples, however, Metabolon was in the process of retiring the GC arm of the platform and the few compounds for which the GC was required did not pass Metabolon's quality standards, therefore there were no reported GC results in the paper and no raw files were provided in the form of mzML files. Thus, GC-MS method has been removed in the current revised manuscript to avoid confusion.

2) For all uploaded LC-MS/MS data files, it included 3 different LC-MS methods: Waters BEH C18 column - acidic condition (method 1); Waters BEH C18 column - basic condition (method 3); Waters BEH Amide column (method 4). However, the method descriptions were NOT matched with the description in main text, “P23 Line 814 in main text: Extracts were divided into five fractions: one for Ultra-performance liquid chromatography tandem mass-spectrometry (UPLC-MS/MS) with positive ion mode electrospray ionization (IMEI); one for (UPLC-MS/MS) with negative IMEI; one for liquid chromatography (LC) polar platform; and one Gas Chromatography (GC)-MS.” I assume the authors discarded the GC-MS data?

Answer: As described above, while GC-MS was utilized for analysis of the samples, Metabolon’s current iteration of the platform had just been validated and deployed for this analysis. The small subset of biochemicals that were contributed from GC-MS did not pass Metabolon’s quality control, thus this data stream did not contribute to any results of the current study. The methods have been revised to reflect this.

3) The details of LC-MS methods were not provided in both *MetaboLights* and manuscript, for example, the elution gradients for each LC-MS method.

Answer: While the details of the chromatographic gradient are not typically included in the files provided from Metabolon, these details have been published in a paper referenced in the original submission (¹; reference 49 of the original manuscript). We have now included the elution gradients for each LC-MS method in **Supplementary Table 22**. Details of MS methods have been also provided in the revised section method and **Supplementary Table 23**. This information has now been included in both the manuscript and *MetaboLights*.

4) The details for metabolite identifications were missing. As a standard reporting protocol, the authors should provide a table with all identified metabolites, and indicated the following parameters:

- Experimental m/z , library m/z ., relative error
- Experimental rt , library rt ., relative error
- MS/MS spectral similarity score
- Positive or negative mode (this information is already provided)

Answer: We appreciate the reviewer’s comments and fully support transparency in the generation of metabolomics data. To this end, Metabolon is actively working with other leaders in the field to frame the standards by which metabolomics data are generated and shared with the scientific community. However, regarding the additional parameters the reviewer suggests, caution must be exercised when there are analytical distinctions that confound the appropriate interpretation of this information. For example, experimental m/z would be slightly different for each sample and each metabolite making that information of limited utility. Further, since our matching is done based on retention index rather than retention time, the latter would be misleading at best. Similar to the issue with experimental m/z , any spectral similarity score would be unique to every metabolite in every sample, thus impractical to include and likely highly confusing. That said, we agree with the reviewer that parameters should be included for the unequivocal identification of metabolites and, in fact, we provide retention index (RI), library m/z and positive or negative mode parameters. The revised manuscript includes this information in **Supplementary Tables 25 and 26** for ventral prostate tissues (VP) samples and **Supplementary Table 27** for serum samples.

2. In addition, I share the same conclusion with Reviewer 2 that the manuscript lacks many important details regarding to the metabolomics experiments, such as sample preparation, metabolite identification etc. It is worthy to note that these experimental details are not the same as the raw data. Even the authors deposited the raw metabolomics data and a part of details to the public database, it is still necessary to provide the detailed experimental descriptions both in the main text and database. Specifically, here are some examples for missing experimental details including but not limited to.

Answer: We thank the reviewer for the observation. The current method section has been significantly improved. Detailed experimental descriptions have now been included in the revised manuscript, supplementary tables, as well as uploaded in *MetaboLights*.

1) P23 Line 808: What were the weights of tissues? What is the recovery standard?

Answer: We apologize that the methods were not clearer, we have now included the required details. Weights are provided in both **Supplementary Tables 25** (raw data), **26** (OrigScale data corrected for inter-day runs). Both tables include the weights and the volume of extraction. Specifically, tissue was weighed on a 4 position analytical scale (1/10th mg) and tissue specimens were soaked overnight in 80% / 20% methanol with recovery standards at a 60 µL : 1mg ratio (due to specimen mass limitations). Recovery standards have been published in the paper referenced above (¹; reference 49 of the original submission) and are now also provided in the revised methods section.

2) P23 Line 814: How much of methanol was added during the preparation?

Answer: Regarding the prostate tissues, see reply above. For serum, 100 µl sample volume was extracted with 500 µl of methanol containing recovery standards. This information is now provided in the revised methods section.

3) P23 Line 820: What were the detailed parameters for UPLC-MS/MS and GC-MS analysis? Please provided the details for “with some modifications” compared to ref 26.

Answer: The details of Metabolon published methods for UPLC-MS/MS are now provided in the revised Methods section and **Supplementary Tables 21-23**. This information reflects the data generation performed for the current study. As discussed above, GC-MS methods have been removed from the current revised version of the manuscript.

4) The authors should add the description of data acquisition in the main text, such as instruments, chromatography columns, solvents, mobile phases etc.

Answer: We thank the reviewer for these comments and agree that data generation is central to how these results should be interpreted. We have now included all these details in the revised methods section.

5) P23 Line 822: What was the meaning of “small volume” in the preparation of pooled sample? Please add the specific volumes.

Answer: 20 µl of each experimental sample was used to make a pooled QC matrix. Method section has been revised accordingly.

6) P23 Line 824: What were included in cocktail of QC standards? It determined how the data quality was checked.

Answer: Details on internal standards are now provided in **Supplementary Table 21**.

7) P23 Line 830: *The authors should claim the used software in peak detection and integration (even it is the commercial software from Metabolon).*

Answer: Metabolon has developed proprietary in-house software that was used in peak detection and integration. Their chemocentric approach can be found in published, cited references below:

- a. DeHaven CD, Evans AM, Dai H, Lawton KA (2010) Organization of GC/MS and LC/MS metabolomics data into chemical libraries. *J Cheminform* 2:9.²
- b. DeHaven CD, Evans AM, Dai H, Lawton KA (2012) Software Techniques for Enabling High-Throughput Analysis of Metabolomics Datasets. *InTech Open*.³
- c. Evans AM, DeHaven CD, Barrett T, Mitchell M, Milgram E, et al. (2009) Integrated, Non-targeted Ultrahigh Performance Liquid Chromatography/Electrospray Ionization Tandem Mass Spectrometry Platform for the Identification and Relative Quantification of the Small-Molecule Complement of Biological Systems. *Anal Chem* 81: 6656-667.⁴

These references have now been included in the revised manuscript. Details relative to peak detection and integration have been also included in the revised methods section.

8) P24 Line 838: *What were the spiked standards to calculate retention index?*

Answer: The standards are a mixture of isotopically labelled compounds including amino acids, fatty acids and other classes as well as some halogenated amino acids. In all cases, the labelling renders the compounds clearly distinct from any endogenous biologically occurring compounds which might complicate their identification and interfere with their use in alignment. Details have been provided in **Supplementary Table 21**.

3. P24 Line 853: *In metabolite identification, the authors said “retention index within 150 RI units of the proposed identification, experimentally detected precursor mass match to the authentic standard within 0.4 m/z and a MS/MS fragmentation spectral score.” Here, a tolerance of 0.4 m/z was too large for the precursor match. For Q-Exactive mass spectrometer, the typical set tolerance should be 5-10 ppm. Again, the authors used both LC-MS and GC-MS which should use two different strategies for identification. Please provide more details for metabolite identification, including the cut-off score.*

Answer: We apologize that the tolerance for the previous iteration of our platform for our mass match to authentic reference standards was included in the manuscript. The revised methods reflect the current methodology for the Metabolon platform that uses a mass tolerance for UPLC-MS analysis of ± 10 ppm. The manuscript has been revised to correct this information. As mentioned above, GC-MS method has been removed.

4. P24 Line 878: *The corrected P value (e.g. FDR) cutoff was set as less than 0.15. Why the authors set such a very loose condition?*

Answer: The acceptable FDR cutoff depends on the goal of the study. The authors feel that the choice of $p < 0.05$ and $FDR < 0.15$ is acceptable in the context of their integrative study. This choice is consistent with several recent publications that exploit metabolic profiling.^{5, 6, 7, 8, 9}

5. P25 Line 883: *How did the authors classified the metabolites? It should add some description or a reference.*

Answer: Biochemical annotations were assigned by PhD-level biochemists at Metabolon, integrating information from literature and public databases (e.g. HMDB). Metabolon regularly reviews biochemical annotations and revised annotation when sufficient biological understanding is available. Since biochemicals may be present in multiple biochemical pathways, they have chosen to annotate biochemicals into the metabolic pathways that will be the most representative of that biochemical. This information has now been included in the revised method section.

6. P4 Line 136: the authors mentioned that “untargeted metabolomics identified 414 metabolites in the prostate”. However, I noticed a note in Supplementary Table S3, “Compound not officially confirmed based on a standard, but confident on its identity.” Therefore, the confidence level of annotations for each metabolite should be clearly given according to the definition of Metabolomics Standards Initiative (MSI). Specifically, the annotation details but not only the result of each metabolite are required to be provided, such as m/z error, RI error, MS/MS database source and MS/MS match score. It would be helpful to evaluate and confirm the accuracy and reliability of metabolite identifications.

Answer: As stated above, we appreciate the reviewer’s request for transparency and wish to comply with the most useful, appropriate data, however, the inclusion of this information for every metabolite would be infeasible and of limited utility. The vast majority of Metabolon identifications are Tier 1, predicated on an authentic chemical reference standard. The asterisk (*) denotes a Tier 2 identification in which no commercially available authentic standard was available. However, the identification was based on spectral and chromatographic similarity to Tier 1 identified compounds and RI, as well as accurate mass is provided for these molecules.

7. A final important comment on study design, it seems that the high-fat diet has a major impact on the lipidome instead of metabolome. For example, in figure 1g, the lipid was a class of metabolites with the most significant change. Similarly, in the supplementary table 3, 53 out of 143 significantly changed metabolites were lipids. Therefore, I would suggest the authors consider an additional lipidomics experiment. It may help to discover more interesting results.

Answer: We thank the reviewer for this suggestion. The authors are indeed currently carrying on these lipidomics experiments. At this time, the authors feel that this analysis is beyond the scope of the current manuscript and the effect of HFD on the lipidome will be in-depth investigated in a separate study. We have included a brief discussion of the importance of MYC-mediated lipid deregulation in prostate cancer and our ongoing lipidomics studies to investigate the interplay between diet-derived fats and MYC-driven *de novo* lipid synthesis in shaping the tumor lipidome and in promoting a more aggressive phenotype.

References

1. Evans AM, *et al.* High Resolution Mass Spectrometry Improves Data Quantity and Quality as Compared to Unit Mass Resolution Mass Spectrometry in High-Throughput Profiling Metabolomics. *Metabolomics* **4**, 132 (2014).
2. Dehaven CD, Evans AM, Dai H, Lawton KA. Organization of GC/MS and LC/MS metabolomics data into chemical libraries. *J Cheminform* **2**, 9 (2010).
3. Dehaven CD, Evans AM, Dai H, Lawton KA. Software Techniques for Enabling High-Throughput Analysis of Metabolomic Datasets. In: *Metabolomics* (ed[^](eds Roessner U). IntechOpen (2012).
4. Evans AM, DeHaven CD, Barrett T, Mitchell M, Milgram E. Integrated, nontargeted ultrahigh performance liquid chromatography/electrospray ionization tandem mass spectrometry platform for the identification and relative quantification of the small-molecule complement of biological systems. *Anal Chem* **81**, 6656-6667 (2009).
5. Cambiaghi A, *et al.* Characterization of a metabolomic profile associated with responsiveness to therapy in the acute phase of septic shock. *Sci Rep* **7**, 9748 (2017).
6. Cruickshank-Quinn CI, *et al.* Metabolomics and transcriptomics pathway approach reveals outcome-specific perturbations in COPD. *Sci Rep* **8**, 17132 (2018).
7. Germain A, Ruppert D, Levine SM, Hanson MR. Metabolic profiling of a myalgic encephalomyelitis/chronic fatigue syndrome discovery cohort reveals disturbances in fatty acid and lipid metabolism. *Mol Biosyst* **13**, 371-379 (2017).
8. Huang J, *et al.* Prospective serum metabolomic profiling of lethal prostate cancer. *Int J Cancer*, (2019).
9. Min HK, Sookoian S, Pirola CJ, Cheng J, Mirshahi F, Sanyal AJ. Metabolic profiling reveals that PNPLA3 induces widespread effects on metabolism beyond triacylglycerol remodeling in Huh-7 hepatoma cells. *Am J Physiol Gastrointest Liver Physiol* **307**, G66-76 (2014).

REVIEWERS' COMMENTS:

Reviewer #6 (Remarks to the Author):

I thank the authors tried their best to provide the metabolomics experimental details as much as possible. Although there are still some missing details (for example, my previous comment 1- (4) and comment 6 for metabolite identification), I guess the author could not obtain the details from the commercial service company. They may claim it is their technology secrete.

At the current situation, I think I have to recommend the acceptance of this manuscript.

But in the future, I strongly suggest the authors not using the same service company any more since they cannot disclose all experimental details. However, these experimental details, particularly for metabolite identifications, are essentially required for most prestigious journals.